# Fast Doubly-Adaptive MCMC to Estimate the Gibbs Partition Function with Weak Mixing Time Bounds

**Shahrzad Haddadan**ⓡ*
Brown University
The Data Science Initiative
shahrzad.haddadan@gmail.com

**Yue Zhuang**ⓡ
Brown University
The Data Science Initiative
yue_zhuang1@brown.edu

**Cyrus Cousins** ⓡ
Brown University
Department of Computer Science
cyrus_cousins@brown.edu

**Eli Upfal**
Brown University
Department of Computer Science
eliezer_upfal@brown.edu

## Abstract

We present a novel method for reducing the computational complexity of rigorously estimating the *partition functions* (normalizing constants) of Gibbs (Boltzmann) distributions, which arise ubiquitously in probabilistic graphical models.

A major obstacle to practical applications of Gibbs distributions is the need to estimate their partition functions. The state of the art in addressing this problem is multi-stage algorithms, which consist of a cooling schedule, and a mean estimator in each step of the schedule. While the cooling schedule in these algorithms is adaptive, the mean estimation computations use MCMC as a black-box to draw approximate samples. We develop a *doubly adaptive* approach, combining the adaptive cooling schedule with an adaptive MCMC mean estimator, whose number of Markov chain steps adapts dynamically to the underlying chain. Through rigorous theoretical analysis, we prove that our method outperforms the state of the art algorithms in several factors: (1) The computational complexity of our method is smaller; (2) Our method is less sensitive to loose bounds on mixing times, an inherent component in these algorithms; and (3) The improvement obtained by our method is particularly significant in the most challenging regime of high-precision estimation. We demonstrate the advantage of our method in experiments run on classic factor graphs, such as voting models and Ising models.

## 1 Introduction

The Gibbs (Boltzmann) distribution is a family of probability distributions of exponential form. First introduced in the context of statistical mechanics [25], Gibbs distributions are now ubiquitous in a variety of other disciplines, such as chemistry [24, 31], economics [1, 54] and machine learning. Gibbs distributions are typically used to model the global state of a system as a function of a collection of interdependent random variables, each representing local states in the system. The dependencies

---

*ⓡ indicates randomized ordering and equal contribution.

35th Conference on Neural Information Processing Systems (NeurIPS 2021).

in the system are modeled by a *Hamiltonian* function, and the probability distribution is inversely proportional to exponent of the Hamiltonian scaled by the *temperature* (see eq. (1) § 1.1).

Gibbs distributions provide potent statistical inference tools in many machine learning applications. They appear in probabilistic graphical models [41, 49, 51], including restricted Boltzmann machines [43, 66], Markov random fields [40, 47], and Bayes networks [32], and are applied in the analysis of images and graphical data [21, 23, 44, 65], topic modeling (LDA) [27, 53, 57, 62], and more [2, 13, 18, 19, 26, 30, 48, 56, 58, 68].

A major obstacle in applying the Gibbs distribution in practice is the need to compute, or estimate, its *partition function* (normalizing constant), henceforth written GPF. The partition function is defined over the Cartesian product of supports of a (typically large) number of variables, making exact computation intractable. Furthermore, due to interdependence of variables, exact sampling is not practically feasible, thus Markov-chain Monte-Carlo (MCMC) solutions for this problem have been extensively studied [6, 22, 29, 34, 37, 39, 42, 52, 63, 67].

Like other MCMC methods, here various heuristics are used. The most well-known heuristics are the *annealed importance sampling* [39, 52, 64] or *convergence diagnostics methods* [10, 11, 20, 61]. Unfortunately, these methods are often error-prone, as their correctness is only proven asymptotically, without rigorous mathematical analysis to bound their estimation error with *finite samples*. In fact, theoretical findings have shown that with no prior knowledge of relevant measures, such as the variance of importance weights in annealed importance sampling, or upper bounds on mixing or relaxation times for diagnostic methods, these methods are either unreliable or computationally intractable (see [52, section 4] or [8, 33]).

On the other hand, theoreticians study this problem by designing Fully Polynomial Randomized Approximation Schemes (FPRAS) (see problem 1). The state of the art FPRAS for estimating the GPF is a multi-stage algorithm involving a sequence of functions at various temperatures, such that the expectation of the product of these functions, or the product of the expectations of said functions, is the GPF. FPRAS's are proven to produce (approximate) solutions w.h.p., but their performance guarantees rely on available upper-bounds on various measures such as variances of estimators or mixing times of Markov chains. In static algorithms, these upper-bounds are given *a priori*, and adaptive[2] algorithms estimate them dynamically, while increasing the sample size until desired properties are mathematically guaranteed. Thus, adaptive algorithms are less sensitive to looseness of known upper-bounds, more robust, often faster, and more easily applied to various settings.

Most of the research on designing FPRAS's for the GPF is focused on designing adaptive algorithms to produce sequences (*cooling schedules*) with minimum length while keeping the variances of estimators small (thus removing the need to have a-priori known bounds on variances). In contrast, the computation of the sequence of mean estimates, which dominates the total computation cost, is done by black-box MCMC estimators, with *a priori* known upper bounds on the mixing times of the chains. These upper bounds are often loose, and improving them for particular models is a challenging active area of research [4, 5, 9, 12, 28, 63]. In order to complement the adaptive cooling schedule and reduce dependence on *a priori* bounds on Markov chains' mixing times, it seems necessary to design an *adaptive* procedure with theoretical guarantees for MCMC-mean estimation.

In this work we develop a *doubly adaptive* FPRAS, combining the adaptive cooling schedule with adaptive MCMC mean estimator that dynamically adapts the number of Markov chain steps to the observed underlying chain. Through rigorous theoretical analysis, we prove that our method outperforms the state of the art algorithms in several factors: (1) The computational complexity of our method is smaller; (2) Our method is less sensitive to loose bounds on mixing times, an inherent component in these algorithms; and (3) The improvement obtained by our method is particularly significant in the most challenging regime of high precision estimates. We demonstrate the advantage of our method in experiments run on classic factor graphs, such as voting and Ising models [5, 7, 15].

## 1.1 Preliminaries and Prior Work

Assume a *sample space* $\Omega$, *Hamiltonian function* $H : \Omega \to \{0\} \cup [1, \infty)$, and *inverse temperature* parameter $\beta \in \mathbb{R}$, referred to as *inverse temperature*. The *Gibbs distribution* on $\Omega$, $H(\cdot)$, and $\beta$ is

---

[2]The usage of the word "adaptive" here refers to algorithms which draw samples progressively and adapt their sample complexity based on empirical estimates until desired conditions are met, as it has been used in [34, 42] (see § 1.1), and should not confused with the work of [60].

then characterized by probability law

$$\forall x \in \Omega : \ \pi_\beta(x) \doteq \frac{1}{Z(\beta)} \exp\big(-\beta H(x)\big) \ . \tag{1}$$

Here $Z(\beta)$ is the *normalizing constant* or *Gibbs partition function* (GPF) of the distribution, with

$$Z(\beta) \doteq \sum_{x \in \Omega} \exp\big(-\beta H(x)\big) \ . \tag{2}$$

Estimating the GPF $Z(\beta)$, is computationally challenging, since typically the size of $\Omega$ is exponential in the number ofvariables, and the values of random terms in the sum have large variance (due to the exponential). The following problem has been extensively studied, and is the focus of this paper.

**Problem 1.** *Given a domain $\Omega$, a Hamiltonian function $H$, and a parameter $\beta$, design a Fully Polynomial Randomized Approximation Scheme (FPRAS) for estimating the partition function $Z(\beta) \doteq \sum_{x \in \Omega} \exp\big(-\beta H(x)\big)$. In other words, for user-supplied $\varepsilon$, the task is to produce an estimate $\hat{Z}(\beta)$, such that with probability at least $1 - \delta$, we have $(1 - \varepsilon)Z(\beta) \leq \hat{Z}(\beta) \leq (1 + \varepsilon)Z(\beta)$, in time polynomial in $1/\varepsilon$, $\ln(1/\delta)$, and all other problem parameters (e.g., the number of vertices in an Ising model, or neurons in an RBM).*

All known scalable solutions to this problem rely on Monte-Carlo Markov-chain (MCMC) methods, and their execution cost is dominated by the total number of Markov chain steps they execute. We therefore follow past work, and analyze our algorithms in terms of number of the Markov chain steps.

**TPA-Based Adaptive Cooling Schedules**   Building on extensive earlier work [6, 22, 67], the current state of the art is due to Huber and Schott [35], with Kolmogorov's sharper analysis [42]. They introduce the *paired product estimator* (PPE), see definition 1.1, and apply the *tootsie-pop algorithm* (TPA) to adaptively compute a near-optimal *cooling schedule*, i.e., a sequence of inverse temperatures $\beta_0 < \beta_1 < \cdots < \beta_{\ell-1} < \beta_\ell$ satisfying $\beta_\ell = \beta$, and that $Z(\beta_0)$ is easy to compute, e.g., $\beta_0 = 0$ is often convenient, since $Z(0) = |\Omega|$. We thus define $Q \doteq Z(\beta)/Z(\beta_0)$ and estimate it using the paired product estimator.

**Definition 1.1** (PPE [34]).  *Assume a cooling schedule $\beta_0, \beta_1, \ldots, \beta_\ell$. For each pair $(\beta_i, \beta_{i+1})$ in the schedule, we define two random variables, $X_i \sim \pi_{\beta_i}$ and $Y_i \sim \pi_{\beta_{i+1}}$, all independent, and we then define $f_{\beta_i, \beta_{i+1}} \doteq \exp\big(-\frac{\beta_{i+1}-\beta_i}{2} H(X_i)\big)$ and $g_{\beta_i, \beta_{i+1}} \doteq \exp\big(\frac{\beta_{i+1}-\beta_i}{2} H(Y_i)\big)$. It is easy to verify that $\mathbb{E}[f_{\beta_i, \beta_{i+1}}] = Z(\frac{\beta_i+\beta_{i+1}}{2})/Z(\beta_i)$, and $\mathbb{E}[g_{\beta_i, \beta_{i+1}}] = Z(\frac{\beta_i+\beta_{i+1}}{2})/Z(\beta_{i+1})$. We then define $F \doteq \prod_{i=1}^k f_{\beta_i, \beta_{i+1}}$, $G \doteq \prod_{i=1}^k g_{\beta_i, \beta_{i+1}}$. Letting $\hat{\mu}$ and $\hat{\nu}$ denote empirical estimates of $\mathbb{E}[F]$ and $\mathbb{E}[G]$, respectively, the* paired product estimator *(PPE) is $\hat{Q} \doteq \hat{\mu}/\hat{\nu}$ .*

Denote by $\mathbb{V}_{\mathrm{rel}}[X] \doteq \mathbb{E}[X^2]/\mathbb{E}[X]^2 - 1 = \mathbb{V}[X]/\mathbb{E}[X]^2$ the *relative variance* of a random variable $X$. *The TPA schedule* [35, 36] is generated by an adaptive algorithm, which, by a proper setting of parameters, outputs a cooling schedule guaranteeing constant $\mathbb{V}_{\mathrm{rel}}[F]$ and $\mathbb{V}_{\mathrm{rel}}[G]$ (see alg. 3 in the supplementary material). Kolmogorov [42] presents a tighter analysis of Huber's TPA method, and proves that with slight modifications (see alg. 4. in the Appendix) the schedule has a shorter length, while preserving constant relative variance for the paired product estimators (see thm. 1.1). In this paper, we use Kolmogorov's algorithm, and we denote it by $\text{TPA}(k, d)$. For completeness, both of Huber's and Kolmogorov's versions of TPA are presented in the Appendix.

We will use the following result in our analysis:

**Theorem 1.1** ([42]).  *Let $H_{\max} \doteq \max_{x \in \Omega} H(x)$, using $\text{TPA}(k, d)$, $k = \Theta(\log H_{\max})$ and $d = 16$ to generate cooling schedule $(\beta_0, \beta_1, \ldots, \beta_\ell)$. W.h.p., we have $\ell = \Theta(\log(Q) \log(H_{\max}))$ and $\mathbb{V}_{\mathrm{rel}}[F]+1 = \prod_{i=1}^\ell (\mathbb{V}_{\mathrm{rel}}[f_{\beta_i, \beta_{i+1}}]+1) = \Theta(1)$ and $\mathbb{V}_{\mathrm{rel}}[G]+1 = \prod_{i=1}^\ell (\mathbb{V}_{\mathrm{rel}}[g_{\beta_i, \beta_{i+1}}]+1) = \Theta(1)$.*

Kolmogorov [42] nearly matches known lower bounds when given *oracle access* to near-independent samples, but leaves open the possibility of better use of the dependent sequence of samples generated by MCMC chains. This fertile ground is ill-explored, since if an approximate sampling oracle draws samples by running a chain for $T$ steps, there is a factor $T$ potential improvement.

**MCMC Mean-Estimator** Huber and Schott [35] assume unit-cost for exact sampling from each $\pi_{\beta_i}$, and Kolmogorov [42] extends their analysis to include the complexity of generating *approximate samples* with standard MCMC processes, assuming *a priori* upper-bounds on their mixing times. The main contribution of our paper is a specialized, adaptive, *multiplicative* MCMC-mean estimator for the TPA-based PPE. Our method is significantly more efficient than using standard black-box MCMC sampling for this problem, thus we improve the best-known method for estimating the GPF.

Let $\mathcal{M}$ be an ergodic Markov chain with state space $S$ and stationary distribution $\pi$. Let $\tau_{\mathrm{mix}}(\varepsilon)$ denote the $\varepsilon$-*mixing time* of $\mathcal{M}$, and define $\tau_{\mathrm{mix}} \doteq \tau_{\mathrm{mix}}(1/4)$. Letting $\lambda$ denote the *second largest absolute eigenvalue* of $\mathcal{M}$'s transition matrix, the *relaxation time* of $\mathcal{M}$ is $\tau_{\mathrm{rx}} \doteq (1 - \lambda)^{-1}$, and it is related to the mixing time $\tau_{\mathrm{mix}}$, by $\left(\tau_{\mathrm{rx}}(\mathcal{M}) - 1\right) \ln(2) \leq \tau_{\mathrm{mix}}(\mathcal{M}) \leq \left\lceil \tau_{\mathrm{rx}}(\mathcal{M}) \ln\left(2/\sqrt{\pi_{\mathrm{min}}}\right) \right\rceil$ [45]. Let $T$ be an upper bound on $\max\{\tau_{\mathrm{rx}}(\mathcal{M}), \tau_{\mathrm{mix}}(\mathcal{M})\}$.

Consider any i.i.d. sampling concentration bound like Chebyshev's, Hoeffding's, or Bernstein's inequalities [50], with, say, sample complexity $m_\varepsilon$. Using MCMC as a black-box sampling tool, we obtain the same precision estimation guarantees, with a computational cost of $m_\varepsilon \cdot \tau_{\mathrm{mix}}(\varepsilon/m_\varepsilon)$, which is equal to $m_\varepsilon \log(m_\varepsilon \cdot \varepsilon^{-1}) \cdot T$ in the absence of exact values for $\tau_{\mathrm{mix}}$.

Other concentration bounds compute the average over the entire trace of a Markov chain, and their complexity is dependent on *known upper-bounds* on the *relaxation time* [14, 38, 46, 50, 55], or function specific mixing time [59]. Note that since $\log(\frac{1}{2\varepsilon})(\tau_{\mathrm{rx}} - 1) \leq \tau_{\mathrm{mix}}(\varepsilon) \leq \log(\frac{1}{\varepsilon \pi_{\mathrm{min}}})\tau_{\mathrm{rx}}$, using these bounds is often more efficient, saving at least $\log(m_\varepsilon)$ steps.

Recently, Cousins et al. [16] introduce a novel Markov chain statistical measure, the *inter-trace variance*. The inter-trace variance depends on both the *function being estimated* and the *dependency structure* between nearby samples in the chain, and unlike the mixing time, it can be efficiently estimated from data. By using progressive sampling, Cousins et al. show an *additive MCMC mean estimator* whose complexity is proved in terms of *inter-trace variance* and they show it it less sensitive to *prior* knowledge of the input parameters, such as relaxation time and trace variance. Unfortunately due to a few technical problems, their result can not directly be used with the TPA method. Thus, in order to obtain a doubly adaptive algorithm for problem 1, we tailor their techniques to our setting, which requires developing new algorithms and analysis tools.

## 1.2 Our Main Contributions

• We present a specialized mean estimator method that significantly improves the state of the art computational complexity of computing the partition function of Gibbs distribution.
• While all rigorous MCMC-based estimates depend on some *a priori* knowledge of the Markov chain properties (such as bounds on its mixing or relaxation time), the complexity of our method is less dependent on these *a priori* bounds, and decays gracefully as they become looser.
• The improvement of our method is particularly significant in the more challenging *high precision* regime, where the goal is to compute estimates with very small multiplicative error.
• Our method improves the computational cost of prior work by replacing standard black-box MCMC mean estimators with an *adaptive MCMC estimator*, specially tailored to this problem.
• The analysis of our method relies on a novel notion of sample variance in a sequence of observations obtained by Markov chains runs, which we term the *relative trace variance*.
• We demonstrate the practicality of our method through experiments on Ising and voting models.

## 2 Algorithms

In this section, we develop two *doubly-adaptive fully polynomial randomized approximation schemes* providing more efficient algorithmic solutions to problem 1. The proof of all of the lemmas and theorems are presented fully in the supplementary material.

**Notation and Setting Parameters** We use the following notation throughout: We use capital letters to denote upper-bounds. e.g., $T$ denotes an upper-bound on $\max(\tau_{\mathrm{mix}}, \tau_{\mathrm{rx}})$, and $\Lambda$ denotes a upper-bound on the second absolute eigenvalue $\lambda$. We use $\mathcal{G}_{H,\beta}$ to denote any Markov chain with Gibbs stationary distribution $\pi_\beta$, eq. (1). Having the Hamiltonian $H$, we denote its maximum and minimum values as $H_{\mathrm{max}}$ and $H_{\mathrm{min}}$, i.e., $H_{\mathrm{max}} \doteq \max_{x \in \Omega}\{H(x)\}$ and $H_{\mathrm{min}} \doteq \min_{x \in \Omega}\{H(x)\}$. Having a schedule $(\beta_0, \beta_1, \ldots, \beta_\ell)$, the paired product estimators $f_{\beta_i,\beta_{i+1}}, g_{\beta_i,\beta_{i+1}}, F = \bigotimes_{i=1}^{\ell} f_{\beta_i,\beta_{i+1}}$ and

$G = \bigotimes_{i=1}^{\ell} g_{\beta_i,\beta_{i+1}}$ are as in definition 1.1. When writing $(\beta_0, \beta_1, \ldots, \beta_\ell) = \text{TPA}(k, d)$, we mean the cooling schedule is obtained from running alg. 4 in the Appendix, and we always set $k = \log H_{\max}$ and $d = 64$, as these parameters are shown to produce a near-optimal schedule w.h.p. [42].

We first introduce a novel MCMC-based *multiplicative* mean estimation procedure RELMEANEST (see alg. 1), and analyze its computational complexity in terms of a new quantity, which we coin *the relative trace variance* (see definition 2.1). RELMEANEST receives as input a Markov chain $\mathcal{M}$, a function $f$, and precision parameters $\varepsilon$ and $\delta$, and it outputs a multiplicative estimate of the expected value of the function w.r.t. the stationary distribution of the Markov chain. For simplicity, we may refer to it as RELMEANEST($\mathcal{M}, f$), leaving out the precision parameters.

Letting $(\beta_0, \beta_1, \ldots, \beta_\ell) = \text{TPA}(k, d)$, we first present PARALLELTRACEGIBBS, in which we invoke both RELMEANEST($\mathcal{G}_{H,\beta_i}, f_{\beta_i,\beta_{i+1}}$) and RELMEANEST($\mathcal{G}_{H,\beta_i}, g_{\beta_i,\beta_{i+1}}$) for each $i = 1, 2, \ldots, \ell - 1$. We then present an often-more-efficient algorithm, SUPERCHAINTRACEGIBBS, which invokes RELMEANEST once each on $F$ and $G$ on a *"super" product chain* (see definition 2.2). We prove correctness of both PARALLELTRACEGIBBS and SUPERCHAINTRACEGIBBS, and bound their complexity in terms of *the relative trace variance* of the estimators. Furthermore, we prove SUPERCHAINTRACEGIBBS improves the computational complexity of the state of the art [42] (thm. 2.4 and corollary 2.6). Both of these algorithms have low dependence on tightness of mixing time: They receive as input an upper-bound on mixing or relaxation time $T$, but we show for $\varepsilon \geq \varepsilon_0$ their computation complexity is dominated by the *true relaxation time* $\tau_{\text{rel}}$ (of each Gibbs chain or the product chain).

## 2.1 Relative trace variance and RELMEANEST

In this section we introduce a new variance notion, the *relative trace variance*, which captures the computational complexity of MCMC-mean estimation with *multiplicative* precision guarantees. The *relative trace variance* depends on both the chain $\mathcal{M}$ and the function $f$, and it generalizes the *relative variance*, defined as $\mathbb{V}_{\text{rel}}[f] \doteq \mathbb{V}[f]/\mathbb{E}[f]^2$, which depends only on $f$, and is used in i.i.d. regimes.

**Definition 2.1** (Relative Trace Variance). *For arbitrary $\tau$, consider a trace of length $\tau$ of a Markov chain $\mathcal{M}$, and a real-valued function $f$. On $\mathcal{M}$, we define the relative trace variance of $f$ as*

$$\text{Reltrv}_{\mathcal{M}}^{\tau}[f] \doteq \frac{\mathbb{E}[\bar{f}(\vec{X}_{1:\tau})^2]}{(\mathbb{E}[\bar{f}(\vec{X}_{1:\tau})])^2} - 1 \ ,$$

*where $\vec{X}_{1:\tau} \doteq X_1, X_2, \ldots, X_\tau$ is a trace of length $\tau$ of $\mathcal{M}$, and $\bar{f}(\vec{X}_{1:\tau}) \doteq (\frac{1}{\tau}) \sum_{i=1}^{\tau} f(X_i)$. We may drop the subscript when the chain is clear from the context.*

The above definition is similar to what Cousins et al. coined as *the inter-trace variance*, denoted by $\text{trv}^{(\tau)}(\mathcal{M}, f)$, which they showed it captures MCMC-mean estimation with *additive* precision guarantees [16]. In fact, the two terms are related as

$$\text{Reltrv}_{\mathcal{M}}^{\tau}[f] = \frac{\text{trv}^{(\tau)}(\mathcal{M}, f)}{(\mathbb{E}[\bar{f}(\vec{X}_{1:\tau})])^2} \ .$$

Note that the two terms are not easily convertible without knowing the *mean*, $\mathbb{E}[\bar{f}(\vec{X}_{1:\tau})]$.

**Lemma 2.1.** *For any $\tau$ we have*

$$\text{Reltrv}_{\mathcal{M}}^{\tau}[f] \leq \mathbb{V}_{\text{rel}}[f] \ . \tag{3}$$

*Furthermore, for $\tau \geq \tau_{\text{rx}}(\mathcal{M})$ we have,*

$$\text{Reltrv}_{\mathcal{M}}^{\tau}[f] = O\left(\frac{\tau_{\text{rx}}(\mathcal{M})}{\tau} \text{Reltrv}_{\mathcal{M}}^{\tau_{\text{rx}}(\mathcal{M})}[f]\right) \ . \tag{4}$$

Lemma 2.1 enables us to compare the computational complexity of our algorithms with the state of the art [42]. In particular, using (3), we show our results improve the state of the art (which is in terms of $\mathbb{V}_{\text{rel}}$), and using (4), we show that for high-precision estimations, the sample complexity of our algorithms only depends on $\tau_{\text{rx}}$, which improves the state of the art (which is in term of $T$).

The relative trace variance is a better analysis tool for estimating the GPF, because, unlike the inter-trace variance, it leads directly to *relative error bounds*, rather than *absolute error bounds*. We now present some definitions which can also be found in standard MCMC textbooks, e.g., [45].

**Definition 2.2** (Product Chain and Tensor Product Function). *Consider $k$ Markov chains $\{\mathcal{M}_i\}_{i=1}^k$ each defined on state space $S_i$ and assume real valued functions $\{f_i : S_i \to \mathbb{R}\}_{i=1}^k$. The* product chain $\mathcal{M}_{1:k}^{\otimes}$ *is defined on the Cartesian product of $S_i$ as follows: at any step $\mathcal{M}_{1:k}^{\otimes}$ chooses $i$ with probability $\omega_i$ (thus $\sum_{i=1}^k \omega_i = 1$), and moves from $(x_1, x_2, \ldots, x_i, \ldots, x_k)$ to $(x_1, x_2, \ldots, y_i, \ldots, x_k)$, with the transition probability of moving from $x_i$ to $y_i$ in $\mathcal{M}_i$. The* tensor product *of $\{f_i\}_{i=1}^k$, denoted by $\bigotimes_{1:k} f_i$, is defined as $\left(\bigotimes_{1:k} f_i\right)(x_1, x_2, \ldots, x_k) = \prod_{i=1}^k f_i(x_i)$.*

**RELMEANEST**  Let $T$ denote an upper bound on the relaxation time of a Markov chain $\mathcal{M}$. RELMEANEST receives $T$, $\mathcal{M}$, $f$ and precision parameters $\varepsilon$ and $\delta$ as input. Before it starts collecting samples, it runs the chain for a *warm start* (§ 2.1 of alg. 1). Starting from a minimum sample size $m^\downarrow$, it runs $\mathcal{M}$ for $T \cdot m^\downarrow$ steps, and collect samples $X_1, X_2, \ldots, X_{T \cdot m^\downarrow}$. It then computes for $j = 1, 2, \ldots, m^\downarrow$, $\bar{f}_j \doteq \sum_{i=(j-1)\cdot T+1}^{j \cdot T} f(X_i)$; using them, it calculates an empirical estimate of the mean, $\hat{\mu}$, and an empirical estimation for the *trace variance* of $\mathcal{M}$ and $f$, $\hat{v}$. Based on these estimates, we derive an upper-bound on the current trace variance $u_i$ and relative error $\hat{\varepsilon}_i^\times$, and check whether is smaller than the user-specified error $\varepsilon$ (lines 18-19). If so, we return the current mean estimate, otherwise we double the sample size and repeat.

---

**Algorithm 1** RELMEANEST

1: **procedure** RELMEANEST
2:   **Input:** Markov chain $\mathcal{M}$, upper-bound on relaxation time $T$, real-valued function $f$ with range $[a, b]$, letting $R = b - a$, multiplicative precision $\varepsilon$, error probability $\delta$.
3:   **Output:** Multiplicative approximation $\hat{\mu}$ of $\mu = \mathbb{E}_\pi[f]$.

4:   $T \leftarrow \left\lceil \frac{1+\Lambda}{1-\Lambda} \ln \sqrt{2} \right\rceil$; $\Lambda' \leftarrow \Lambda^T$          ▷ Choose $T$ to be an upperbound on relaxation time

5:   $I \leftarrow 1 \vee \left\lfloor \log_2 \left( \frac{bR}{2a^2} \cdot \frac{(1-\varepsilon)^2}{(1+\varepsilon)\varepsilon} \right) \right\rfloor$; $\alpha \leftarrow \frac{(1+\Lambda')R \ln \frac{3I}{\delta}(1+\varepsilon)}{(1-\Lambda')b\varepsilon}$; $m_0 \leftarrow 0$          ▷ Initialize *sampling schedule*

6:   $T_{\text{unif}} \leftarrow \left\lceil T \cdot \ln(1/\pi_{\min}) \right\rceil$; $(\vec{X}_{0,1}, \vec{X}_{0,2}) \leftarrow \mathcal{M}^{T_{\text{unif}}}(\perp)$          ▷ Warm-start two chains for $T_{\text{unif}}$ steps from arbitrary $\perp \in \Omega$
7:   **for** $i \in 1, 2, \ldots, I$ **do**
8:     $m_i \leftarrow \lceil \alpha r^i \rceil$          ▷ Total sample count at iteration $i$; $r$ is the geometric ratio (constant, usually 2) size
9:     **for** $j \in (m_{i-1} + 1), \ldots, m_i$ **do**
10:       $(\vec{X}_{j,1}, \vec{X}_{j,2}) \leftarrow (T \text{ steps of } \mathcal{M} \text{ starting at } \vec{X}_{j-1,1}, \vec{X}_{j-1,2})$          ▷ Run two independent copies of $\mathcal{M}$ for $T$ steps
11:       $\bar{f}(\vec{X}_{j,1}) \leftarrow \frac{1}{T} \sum_{t=1}^T f(\vec{X}_{j,1}(t))$; $\bar{f}(\vec{X}_{j,2}) \leftarrow \frac{1}{T} \sum_{t=1}^T f(\vec{X}_{j,2}(t))$          ▷ Average $f$ over $T$-traces
12:     **end for**
13:     $\hat{\boldsymbol{\mu}}_i \leftarrow \frac{1}{2m_i} \sum_{i=1}^{m_i} (f(\vec{X}_{j,1}) + f(\vec{X}_{j,2}))$; $\hat{v}_i \leftarrow \frac{1}{2m_i} \sum_{i=1}^{m_i} ((f(\vec{X}_{j,1}) - f(\vec{X}_{j,2}))^2$          ▷ Compute empirical mean; trace variance

14:     $u_i \leftarrow \hat{v}_i + \frac{(11 + \sqrt{21})(1 + \Lambda'/\sqrt{21})R^2 \ln \frac{3I}{\delta}}{(1-\Lambda')m_i} + \sqrt{\frac{(1+\Lambda')R^2 \hat{v}_i \ln \frac{3I}{\delta}}{(1-\Lambda')m_i}}$          ▷ Variance upper bound

15:     $\hat{\varepsilon}_i^+ \leftarrow \frac{10R \ln \frac{3I}{\delta}}{(1-\Lambda')m_i} + \sqrt{\frac{(1+\Lambda')u_i \ln \frac{3I}{\delta}}{(1-\Lambda')m_i}}$          ▷ Apply Bernstein bound

16:     $\hat{\boldsymbol{\mu}}_i^\times \leftarrow \frac{(\hat{\boldsymbol{\mu}}_i - \hat{\varepsilon}_i^+) \vee a + (\hat{\boldsymbol{\mu}}_i + \hat{\varepsilon}_i^+) \wedge b}{2}$          ▷ Optimal mean estimate

17:     $\hat{\varepsilon}_i^\times \leftarrow \frac{((\hat{\boldsymbol{\mu}}_i + \hat{\varepsilon}_i^+) \wedge b - (\hat{\boldsymbol{\mu}}_i - \hat{\varepsilon}_i^+) \vee a}{2\hat{\boldsymbol{\mu}}_i^\times}$          ▷ Empirical relative error bound

18:     **if** $(i = I) \vee (\hat{\varepsilon}_i^\times \leq \varepsilon)$ **then**          ▷ Terminate if accuracy guarantee is met
19:       **return** $\hat{\boldsymbol{\mu}}_i^\times$
20:     **end if**
21:   **end for**
22: **end procedure**

---

The following theorem, shows the correctness of RELMEANEST and bounds its complexity.

**Theorem 2.2** (Efficiency and Correctness of RELMEANEST). *With probability at least $1 - \delta$, RELMEANEST will output $\hat{\mu}$ satisfying $(1 - \varepsilon)\hat{\mu} \leq \mu \leq (1 + \varepsilon)\hat{\mu}$. Furthermore, with probability at least $1 - \frac{\delta}{3I}$, the total Markov chain steps of RELMEANEST, $\hat{m}$, obeys*

$$\hat{m} \in \mathcal{O}\left( \ln\left( \frac{\ln \frac{b}{a\varepsilon}}{\delta} \right) \left( \frac{T \cdot R}{\mu\varepsilon} + \frac{\tau_{\text{rx}} \text{Reltrv}^{\tau_{\text{rx}}}}{\varepsilon^2} \right) \right). \tag{5}$$

## 2.2 Doubly adaptive algorithms: SUPERCHAINTRACEGIBBS and PARALLELTRACEGIBBS

Let $(\beta_0, \beta_1, \ldots, \beta_\ell) = \text{TPA}(k, d)$, and consider a family of Gibbs chains $\mathcal{G}_{H,\beta_i}$, each corresponding to some $\beta_i$, and the paired product estimators $F = \bigotimes_{i=1}^\ell f_{\beta_i, \beta_{i+1}}$, $G = \bigotimes_{i=1}^\ell g_{\beta_i, \beta_{i+1}}$. The TPA method is designed to ensure $\mathbb{V}_{\text{rel}}$ of the estimators are bounded, which can be employed by concentration bounds (e.g., Chebyshev's bound) to guarantee the multiplicative error is bounded with high probability for a given sample size.

In order to generalize the same machinery for samples generated from a Markov chain using RELMEANEST, we need to bound the two terms appearing in eq. (5), which dominate the computational complexity of RELMEANEST. We refer to the first term, $T \cdot R/\mu$, as the *range term*, and to the term $\tau_{\text{rx}}\text{Reltrv}^{\tau_{\text{rx}}}$ as the *trace variance term*. Note that as $\varepsilon$ becomes smaller, the *trace variance* term dominates the sample complexity of RELMEANEST, thus dependence on loose bounds $T$ and $R$ is dominated by dependence on *true and a priori unknown* values $\tau_{\text{rx}}$ and $\text{Reltrv}^{\tau_{\text{rx}}}$.

In order to ensure that the ranges of estimators are small, we prove that the length of each inverse-temperature interval in the TPA schedule is w.h.p. small. Having a schedule $(\beta_0, \beta_1, \ldots, \beta_\ell)$ we define and use the following notation: for $0 \leq i \leq \ell - 1$, *interval length* $\Delta_i \doteq \beta_{i+1} - \beta_i$, *maximum interval length* $\Delta_{\max} \doteq \max_i \Delta_i$, and *total length* $\Delta \doteq \beta_\ell - \beta_0$.

**Lemma 2.3.** *Let $z(\beta) \doteq \ln\big(Z(\beta)\big)$, and let $\beta_i$, $\beta_{i+1}$ be two consecutive points generated by* TPA$(k, d)$. *For arbitrary $\varepsilon \geq 0$, we have:*

1. $\mathbb{P}(z(\beta_i) - z(\beta_{i+1}) \leq \varepsilon) \geq (1 - \exp(-\varepsilon k/d))^d$.
2. $\mathbb{P}\big(\Delta_i \geq \varepsilon/\mathbb{E}[H(x)]\big) \leq d\exp(-\varepsilon k/d)$, *where $\mathbb{E}[H(x)]$ is taken w.r.t. $x \sim \pi_{\beta_{i+1}}$.*

**SUPERCHAINTRACEGIBBS** Let $\mathcal{G}^{\otimes}$ the product of $\mathcal{G}_{H,\beta_i}$s with uniform weights i.e., $\omega_i = \frac{1}{\ell}, \forall i$ (see definition 2.2). SUPERCHAINTRACEGIBBS calls RELMEANEST$(\mathcal{G}^{\otimes}, F)$ and RELMEANEST$(\mathcal{G}^{\otimes}, G)$, with appropriate parameters, and simply outputs the ratio of the two estimates (see alg. 2, left).

---

**Algorithm 2** SUPERCHAINTRACEGIBBS and PARALLELTRACEGIBBS

```
 1: procedure SUPERCHAINTRACEGIBBS(...)          16: procedure PARALLELTRACEGIBBS(...)
 2:   (β₀, β₁, ..., β_ℓ) ← TPA(k, d)ᵃ            17:   (β₀, β₁, ..., β_ℓ) = TPA(k, d)
```

3: $\varepsilon' \leftarrow \frac{\varepsilon}{2+\varepsilon}; \delta' \leftarrow \frac{\delta}{2}$

18: $\varepsilon' \leftarrow \frac{\sqrt[\ell]{1+\varepsilon}-1}{\sqrt[\ell]{1+\varepsilon}+1}; \delta' \leftarrow \frac{\delta}{2\ell}$

4: **for** $i \in 1, 2, \ldots, \ell$ **do**

19: **for** $i \in 1, 2, \ldots \ell$ **do**

5: $\quad f_i(x) \doteq \exp(-\frac{\beta_{i+1}-\beta_i}{2}H(x))$

20: $\quad f_i(x) \doteq \exp(-\frac{\beta_{i+1}-\beta_i}{2}H(x))$

6: $\quad g_i(x) \doteq \exp(\frac{\beta_i-\beta_{i-1}}{2}H(x))$

21: $\quad g_{i-1}(x) \doteq \exp(\frac{\beta_i-\beta_{i-1}}{2}H(x))$

7: **end for**

22: $\quad R_f \leftarrow \exp(-\frac{\beta_{i+1}-\beta_i}{2}H_{\min}) - \exp(-\frac{\beta_{i+1}-\beta_i}{2}H_{\max})$

8: $\quad F \doteq \bigotimes_{i=1}^\ell f_i; G \doteq \bigotimes_{i=1}^\ell g_i$

23: $\quad R_g \leftarrow \exp(\frac{\beta_{i+1}-\beta_i}{2}H_{\max}) - \exp(\frac{\beta_{i+1}-\beta_i}{2}H_{\min})$

9: $\quad \mathcal{G}^{\otimes} \leftarrow \bigotimes_{i=1}^\ell \mathcal{G}_{H,\beta_i}$, with $\omega_i = \frac{1}{\ell}, \forall i$

24: $\quad \hat{\mu}_i \leftarrow$ RELMEANEST$(\mathcal{G}_i, R_f, T_i, f_i, \varepsilon', \delta')$

10: $\quad R_f \leftarrow \exp(-\frac{\beta-\beta_0}{2}H_{\min}) - \exp(-\frac{\beta-\beta_0}{2}H_{\max})$

25: $\quad \hat{\nu}_i \leftarrow$ RELMEANEST$(\mathcal{G}_i, R_g, T_i, g_i, \varepsilon', \delta')$

11: $\quad R_g \leftarrow \exp(\frac{\beta-\beta_0}{2}H_{\max}) - \exp(\frac{\beta-\beta_0}{2}H_{\min})$

26: **end for**

12: $\quad \hat{\mu} \leftarrow$ RELMEANEST$(\mathcal{G}^{\otimes}, R_f, T, F, \varepsilon', \delta')$

27: **return** $\hat{Z} \leftarrow \prod_{i=1}^\ell \frac{\hat{\nu}_i}{\hat{\mu}_i}$

13: $\quad \hat{\nu} \leftarrow$ RELMEANEST$(\mathcal{G}^{\otimes}, R_g, T, G, \varepsilon', \delta')$

28: **end procedure**

14: **return** $\hat{Z} \leftarrow \frac{\hat{\nu}}{\hat{\mu}}$

15: **end procedure**

---

$^a k = \Theta(\log H_{\max})$ and $d = 64$ as in [42]

---

We now show the correctness and efficiency of SUPERCHAINTRACEGIBBS. Let $\tau_{\text{prx}}$ denote $\mathcal{G}^{\otimes}$'s true (and unknown) relaxation time and $T$ a known upper-bound on it ($T \geq \tau_{\text{prx}}$), $\varepsilon$ and $\delta$ are user specified precision parameters. For simlicity of presentation we use the following notation to refer to relative ranges: $\text{relR} = \text{Range}(F)/\mu + \text{Range}(G)/\nu$, where $\mu = \mathbb{E}[F]$ and $\nu = \mathbb{E}[G]$.

**Theorem 2.4.** *With probability at least $1 - \delta$, it holds that the total number $\hat{m}$ of Markov chain steps taken by* SUPERCHAINTRACEGIBBS *is upper-bounded by*

$$\tilde{\mathcal{O}}\left(\ln\left(\frac{1}{\delta}\right)\left(\frac{T \cdot \text{relR}}{\varepsilon} + \frac{\tau_{\text{prx}} \cdot \big(\text{Reltrv}_{\mathcal{G}^{\otimes}}^{\tau_{\text{prx}}}(F) + \text{Reltrv}_{\mathcal{G}^{\otimes}}^{\tau_{\text{prx}}}(G)\big)}{\varepsilon^2}\right)\right).$$

**Lemma 2.5.** *Defining $\alpha_1 = \sqrt{\frac{Z(\beta_0)}{Z(\beta_0-\Delta_{\max})}}$, we have:* $\quad \frac{\text{Range}(F)}{\mu} \leq \alpha_1 \sqrt{\frac{Q}{\exp(\Delta H_{\min})}}$ *and* $\frac{\text{Range}(G)}{\nu} \leq \alpha_1 \sqrt{\frac{\exp(\Delta H_{\max})}{Q}}$.

| PARALLELTRACEGIBBS | SUPERCHAINTRACEGIBBS | TPA + PPE [42] |
|---|---|---|
| $\ell^2 \sum_{i=1}^{\ell} \tau_i \left( \mathrm{Reltrv}_{\mathcal{G}_{H,\beta_i}}^{\tau_i}(f_i) + \right.$ $\left. \mathrm{Reltrv}_{\mathcal{G}_{H,\beta_i}}^{\tau_i}(g_i) \right)$ | $\tau_{\mathrm{prx}} \left( \mathrm{Reltrv}^{\tau_{\mathrm{prx}}}[F] + \mathrm{Reltrv}^{\tau_{\mathrm{prx}}}[G] \right)$ $= O\left( \ell \max\{\tau_i\}_{i=1:\ell} \right)$ | $\ln \frac{q \ln H_{\max}}{\varepsilon} \sum_{i=1}^{\ell} T_i \cdot \left( \mathbb{V}_{\mathrm{rel}}(F) + \mathbb{V}_{\mathrm{rel}}(G) \right)$ $= O\left( \ln \frac{q \ln H_{\max}}{\varepsilon} \sum_{i=1}^{\ell} T_i \right)$ |

Table 1: Comparison of the number of Markov chain steps, when $\varepsilon$ is adequately small. In all columns, a multiplicative factor of $1/\varepsilon^2$ is omitted to ease presentation, and $q = \ln Q$. Note that computational complexity of both PARALLELTRACEGIBBS and SUPERCHAINTRACEGIBBS only depends on true relaxation times, denoted by $\tau_i$, and the TPA + PPE method's complexity is dependent on their upper bounds, denoted by $T_i$.

Using lemma 2.5 and thm. 2.4, we identify $\varepsilon_0$ such that for $\varepsilon \le \varepsilon_0$ the trace variance term in will the dominate computational complexity of SUPERCHAINTRACEGIBBS. In order to make a fair comparison with the state of the art [42] we employ eq. (3) of lemma 2.1. Finally we use thm. 1.1 and conclude:

**Corollary 2.6.** *Let $\alpha_1$ be as in lemma 2.5, $\tau_{\max} \doteq \max_i \tau_i$ and $\varepsilon_0 \doteq (\tau_{\mathrm{prx}}/T) \cdot \left( \sqrt{\frac{\exp(\Delta H_{\min})}{Q}} + \sqrt{\frac{Q}{\exp(\Delta H_{\max})}} \right) \cdot \alpha_1$. When $\varepsilon \le \varepsilon_0$, the number of Markov chain steps of* SUPERCHAINTRACEGIBBS *is dominated by $\tilde{O}(\ell \tau_{\max})$.*

**PARALLELTRACEGIBBS** For $i = 1, 2, \ldots, \ell - 1$, PARALLELTRACEGIBBS (alg. 2, right) runs RELMEANEST$(\mathcal{G}_{H,\beta_i}, f_{\beta_i,\beta_{i+1}})$ and RELMEANEST$(\mathcal{G}_{H,\beta_i}, g_{\beta_i,\beta_{i+1}})$ independently. We show the computational complexity of PARALLELTRACEGIBBS in thm. 2.7.

For $i = 1, 2, \ldots, \ell$, assume $\tau_i$ is the true (unknown) relaxation time of $\mathcal{G}_{H,\beta_i}$ and $T_i$ is a known bound on it. For simplicity of presentation we use the following notations: $\mathrm{relR}_i \doteq \mathrm{Range}(f_{\beta_i,\beta_{i+1}})/\mu_i + \mathrm{Range}(g_{\beta_{i-1},\beta_i})/\nu_i$, where $\mu_i = \mathbb{E}(f_{\beta_i,\beta_{i+1}})$ and $\nu_i = \mathbb{E}(g_{\beta_i,\beta_{i+1}})$.

**Theorem 2.7** (Efficiency of PARALLELTRACEGIBBS). *With probability at least $1 - \delta$, it holds that the total number $\hat{m}$ of Markov chain steps taken by* PARALLELTRACEGIBBS *is upper-bounded by*

$$\tilde{\mathcal{O}}\left( \log\left(\frac{\ell}{\delta}\right) \sum_{i=1}^{\ell} \left( \frac{\ell \cdot T_i \cdot \mathrm{relR}_i}{\varepsilon} + \frac{\ell^2}{\varepsilon^2} \tau_i \cdot \left( \mathrm{Reltrv}_{\mathcal{G}_{H,\beta_i}}^{\tau_i}(f_{\beta_i,\beta_{i+1}}) + \mathrm{Reltrv}_{\mathcal{G}_{H,\beta_i}}^{\tau_i}(g_{\beta_{i-1},\beta_i}) \right) \right) \right).$$

*Furthermore, for all $1 \le i \le \ell$, $\mathrm{Range}(f_{\beta_i,\beta_{i+1}})/\mu_i \le \ell^{1/\log(n)}$ and $\mathrm{Range}(g_{\beta_{i-1},\beta_i})/\nu_i \le \ell^{\alpha_0(i)/\log n}$, where $\alpha_0(i) = (H_{\max}/2\mathbb{E}[H(x)]) - 1$, for $x \sim \pi_{\beta_i}$.*

PARALLELTRACEGIBBS and SUPERCHAINTRACEGIBBS make different computational complexity tradeoffs. PARALLELTRACEGIBBS is usually slower than SUPERCHAINTRACEGIBBS, because in each iteration $i = 1, 2, \ldots, \ell$, the mean estimator must acquire a higher-precision estimate so that *all estimators together* achieve an $\varepsilon$-$\delta$ relative-error guarantee. Relaxation times (true values and their upper-bounds) appear *in a sum* in the complexity of PARALLELTRACEGIBBS, whereas they appear *in a maximum* in SUPERCHAINTRACEGIBBS ($\sum_{i=1}^{\ell} \tau_i$ vs. $\max_{i=1,\ldots,\ell} \tau_i$). Furthermore, dominance of the trace variance terms in both of these algorithms occur at different values of $\varepsilon$. A comparison of the complexity of these algorithms, in the high-precision regime, with Kolmogorov's TPA + PPE (which uses MCMC as a black box) is presented in table 1.

## 3 Experimental Results

In this section we report our experiment results, comparing the performance of the two versions of our *doubly adaptive* method (alg. 2), to the performance of the state of the art algorithm in [42].

**Setup.** We run the experiments using the single site Gibbs sampler (known also as the Glauber dynamics) on two different factor graph models:

**(A) The Ising model on 2D lattices.** Having a 2-dimension lattice of size $n \times n$, the Hamiltonian is defined on $n^2$ random variables having values $\pm 1$ and their dependency is represented by the Hamiltonian: $H(x) = -\sum_{(i,j) \in E} \mathbb{1}(x(i) = x(j))$. We run the algorithms on lattices of sizes $2 \times 2$,

$3 \times 3$, $4 \times 4$, and $6 \times 6$. For each lattice, the parameter $\beta \geq 0$ is chosen below the critical inverse temperature at which it undergoes a phase transition. We use known mixing time bounds for high temperature Ising models [3] (see fig. 1 and A.6. of supplementary material).

**(B) The logical voting model.** For a parameter $n$, we have $2n + 1$ random variables: the query variable $Q \in \{-1, 1\}$, and the voter variables $T_1, T_2, \ldots, T_n$ and $F_1, F_2, \ldots, F_n$ all in $\{0, 1\}$. The factors have $2n + 1$ weights, $\omega, \omega_{T_i}, \omega_{F_i}, i = 1, \ldots, n$. The Hamiltonian is:

$$H(Q, T, F) = \omega Q \max_i T_i - \omega Q \max_i F_i + \sum_{i=1}^n \omega_{T_i} T_i + \sum_{i=1}^n \omega_{F_i} F_i \text{ , where } \omega, \omega_{T_i}, \omega_{F_i} \in [-1, 1]$$

The parameters are reported in fig. 2. We follow De Sa et al. [17] and use *hierarchy width* to derive upper bounds on mixing times. To make a fair comparison, we always run the TPA algorithms once, and with the parameters given in [42]. At each iteration of RELMEANEST, the sample size is extended with geometric ratio 1.1 (see alg. 1 line 8). All code is available at `https://github.com/zysophia/Doubly_Adaptive_MCMC`.

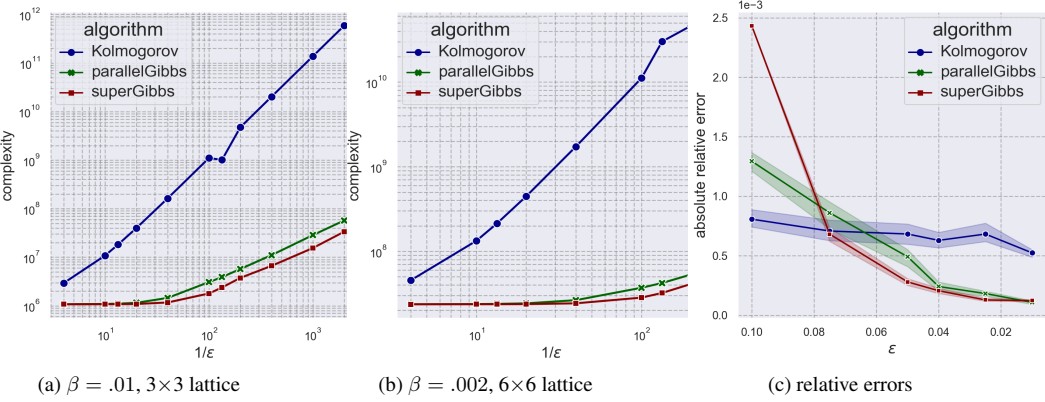

(a) $\beta = .01$, $3 \times 3$ lattice      (b) $\beta = .002$, $6 \times 6$ lattice      (c) relative errors

Figure 1: Comparison of sample complexity and precision $\frac{1}{\varepsilon}$ on Ising models. See also the A.6. of the supplementary material

**Results:** Our experiments demonstrate the practical advantages of our *doubly adaptive method*, validating our theoretical analysis.

(1) We first compare the complexity of our algorithms to Kolmogorov's algorithm. Our experiments show the superiority of both versions of our methods on different models and various sets of parameters. Figure 1 demonstrates the superiority of our methods on the Ising model for various sets of parameters, and in figs. 2a and 2c for the voting model, when $\varepsilon$ is fixed and $Z(\beta)$ is varying (fig. 2c), and when $Z$ is fixed and $\varepsilon$ is varying (fig. 2a). All of these hold while the precision of our algorithms beats [42] as $\varepsilon \to 0$ (fig. 1c).

(2) To demonstrate the advantage of using the relative *trace* variance, in contrast to the relative variance, we run both of our algorithms using a simpler mean estimator which only uses progressive sampling, and we compare the results. This is done by setting $T \leftarrow 1$ in line 4 of RELMEANEST. In Figure 2b, we show the effectiveness of *trace averaging*, since both SUPERCHAINTRACEGIBBS and PARALLELTRACEGIBBS beat their simplified versions ($T \leftarrow 1$) after $1/\varepsilon$ passes a certain threshold. This is consistent for different parameters of the voting model.

(3) Comparing the performance of SUPERCHAINTRACEGIBBS and PARALLELTRACEGIBBS, we observe that in all of our experiments SUPERCHAINTRACEGIBBS has better performance than PARALLELTRACEGIBBS. In fig. 2b, we show the trace variance term PARALLELTRACEGIBBS becomes dominant earlier as $1/\varepsilon$ grows, thus it performs better in this perspective. This is consistent with our theoretical findings, because the ranges of estimators in PARALLELTRACEGIBBS are smaller than the ranges used in SUPERCHAINTRACEGIBBS.

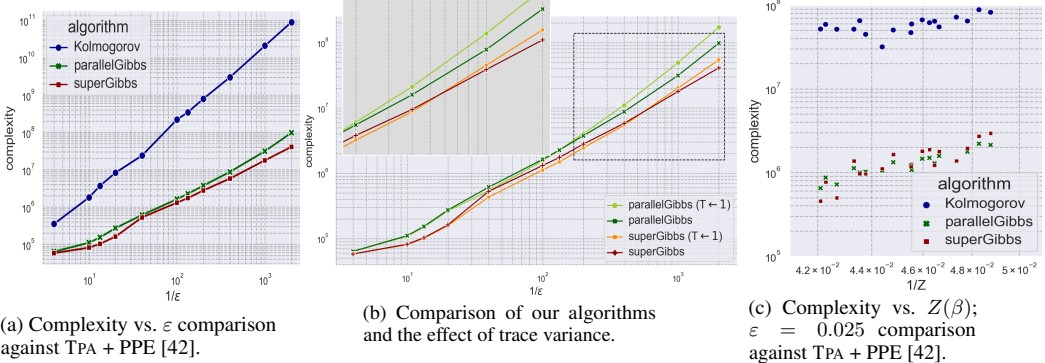

(a) Complexity vs. $\varepsilon$ comparison against TPA + PPE [42].

(b) Comparison of our algorithms and the effect of trace variance.

(c) Complexity vs. $Z(\beta)$; $\varepsilon = 0.025$ comparison against TPA + PPE [42].

Figure 2: Experiments on voting models. In (a) and (b) the parameters are $\beta = 0.1, n = 3, \omega = 0.9, \omega_T = \langle 0.2, 0.5, 0.1 \rangle$ and $\omega_F = -\langle 0.8, 0.2, 0.9 \rangle$. In (c), we have $n = 5$, and the weights and $\beta$ are picked randomly to generate models with various values of $Z(\beta)$.

## 4 Conclusions: advantages and limitations of proposed algorithms

We develop a doubly-adaptive MCMC-based estimator for the partition function of Gibbs distributions, which resolves a major impediment of prior methods that use MCMC as a black-box sampler. We show, both theoretically and experimentally, that our method requires substantially fewer MCMC steps than the state-of-the-art method. The better performance is due to several factors, which all stem from the use of an *adaptive MCMC mean estimator* instead of a standard "black-box" MCMC estimate. The complexity of the adaptive MCMC process depends on the (smaller) *trace*, rather than *stationary*, relative variances, and on *relaxation times* instead of *mixing times*. It is also less sensitive to weak upper-bounds on mixing and relaxation times.

In particular, Kolmogorov's method requires $\Theta(\ell/\varepsilon^2)$ approximately independent samples, where $\ell$ is the length of cooling schedule. This requires tight convergence (total variance distance of $O(\varepsilon^2/\ell)$ from stationary) for each sample, which adds a multiplicative $\ln \frac{\ell}{\varepsilon^2}$, with $\ell = \Theta(\ln Q \ln H_{\max})$, to its complexity (see column 3 of table 1 and [42], theorem 9). In contrast, our doubly adaptive method *only* depends on relaxation times, which do not depend on $\varepsilon$.

**Limitations.** While significantly improving the state of the art, our methods suffer from a several limitations. In SUPERCHAINTRACEGIBBS, the major limitation is the dependence on the *relative ranges* of $F$ and $G$, which can be large, especially when the Hamiltonian range is large. Another issue is that the product chain's mixing time is dominated by $\ell \max\{\tau_i\}_{i=1}^{\ell}$, as opposed to $\sum_{i=1}^{\ell} \tau_i$. While PARALLELTRACEGIBBS circumvents these issues by estimating each factor of the telescoping product independently, it fails to beat SUPERCHAINTRACEGIBBS's efficiency in general, due both to the union bound and the higher-precision guarantees required for each subproblem. Improving performance further will likely require new estimators with smaller ranges and relative trace variances.

**Statement of Broader Impact.** While probabilistic graphical models as other machine learning methods that rely on MCMC estimations continue to grow in importance and popularity. But running the MCMC to theoretical convergence guarantees is often prohibitively expensive, while running it to *apparent convergence* is methodologically unsound, particularly in the modern context, where public confidence in machine learning systems is continuously eroded by ethical, accuracy, and safety failures. Our work attempts to bridge the gap between the definite, elegant and theoretically sound analytic methods, and efficiency-focused practical utility, as we seek to reduce *proof-burden*, while maintaining theoretical guarantees of accuracy, with adaptive methods that bound efficiency in terms of (potentially unknown) convergence rate metrics and variances.

**Acknowledgements.** Shahrzad Haddadan is supported by NSF Award CCF-1740741. Cyrus Cousins and Eli Upfal are supported by NSF grant RI-1813444 and DARPA/AFRL grant FA8750. The authors are thankful to anonymous reviewers of NeurIPS 2021 for several valuable inputs.

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
