# A   Appendix

## A.1   Algorithms used in the literature

### A.1.1   The TPA method [4, 6]

We refer to Huber and Schott's algorithm as the original TPA, and Kolmogorov's, which is used in our algorithms and referred to as $\text{TPA}(k, d)$ in the main manuscript, as the TPA method.

**Algorithm 3** THE ORIGINAL TPA-METHOD [4]

1: **output** a schedule $(\beta_1, \ldots, \beta_l)$ of values in the interval $[\beta_{\min}, \beta_{\max}]$.
2: $\beta_0 \leftarrow \beta_{\min}$
3: **for** $i = 0 : \infty$ **do**
4:     sample $X \sim \pi_{\beta_i}$ draw $U \in [0,1]$ uniformly, $\beta_{i+1} = \beta_i - \log U / H(X)$ (or $+\infty$ if $H(X) = 0$.)
5:     **if** $\beta_{i+1} \notin [\beta_{\min}, \beta_{\max}]$ **then** Terminate
6:     **end if**
7: **end for**

---

**Algorithm 4** TPA-METHOD [6]

1: **input** integers $k$ and $d$
2: **output** a schedule $(\beta_0, \beta_1, \ldots, \beta_l)$ of values in the interval $[\beta_{\min}, \beta_{\max}]$.
3: **for** $i = 1 : k$ **do**
4:     $\mathcal{B}_i \leftarrow$ THE ORIGINAL TPA-METHOD().
5:     let $\mathcal{B} \leftarrow \mathcal{B} \cup \mathcal{B}_i$
6: **end for**
7: sort $\mathcal{B}$, keep one sample uniformly from the initial $d$ elements, and keep every $d$th successive value in the remaining sequence.
8: add $\beta_{\min}$ and $\beta_{\max}$ to $\mathcal{B}$ **return** $\mathcal{B}$

---

### A.1.2 Single site Gibbs sampler (Glauber dynamics chain)

Consider $\beta$ and $H$ defined as above. Let $X = (X_1, X_2, \ldots, X_n)$ be the set of all variables in the Gibbs distribution with inverse temperature $\beta$ and Hamiltonian $H$, thus, the domain of $H$ is $\Omega = \Omega_1 \times \Omega_2 \times \ldots \Omega_n$, and each $\Omega_i$ is the range of random variable $X_i$. At each time step $t$, assume the current state is $x^{(t)} = (x_1, x_2, \ldots, x_n)$. Take $i \sim 1, \ldots, n$ uniformly at random. Sample $y$ from the following distribution:

$$\pi_\beta(y | x_{-i}^{(t)}) = \frac{\exp(-\beta H(x^{(t)}; x_i \leftarrow y))}{\sum_{\omega \in \Omega_i} \exp(-\beta H(x^{(t)}; x_i \leftarrow \omega))} \quad , \tag{1}$$

where for an arbitrary $\omega \in \Omega_i$ we define $(x^{(t)}; x_i \leftarrow \omega)$ be the vector in which all the elements except the $i$th element are equal to $x_i$ and the $i$th element is replaced with $\omega$.

In other words, for any arbitrary vectors $x^{(t)}$ and $x^{(t+1)}$, the transition probability is:

$$\mathcal{G}_{H,\beta}(x^{(t)}, x^{(t+1)}) = \begin{cases} (1/n) \pi_\beta(y | x_{-i}^{(t)}), & \exists y, i \text{ such that } x_i \neq y \text{ and } x^{(t+1)} = (x^{(t)}; x_i \leftarrow y), \\ \sum_{i=1}^n (1/n) \pi_\beta(x_i | x_{-i}^{(t)}) & \text{if } x^{(t)} = x^{(t+1)}, \\ 0 & \text{otherwise .} \end{cases}$$

### A.2 Missing proofs: TPA and relative trace variance properties

**Lemma A.1.** *Let* $z(\beta) \doteq \log(Z(\beta))$, *$d$ and $k$ the parameters of the* TPA *method, and $\beta_i$ and $\beta_{i+1}$ two consecutive points generated by* TPA$(k, d)$, *we have:*

*1. For any $\varepsilon \geq 0$, we have* $\mathbb{P}(z(\beta_j) - z(\beta_{j+1}) \leq \varepsilon) \geq (1 - \exp(-\varepsilon k/d))^d \simeq 1 - d \exp(-\varepsilon k/d)$,
*2. For any $\varepsilon \geq 0$,* $\mathbb{P}\left(\Delta_i \geq \varepsilon / \mathbb{E}[H(x)]\right) \leq d \exp(-\varepsilon k/d)$, *where the expectation of $H(x)$ is taken with respect to distribution $x \sim \pi_{\beta_{i+1}}$.*

*Proof of lemma A.1.* Note that TPA$(k, d)$ of [6] consists of $k$ parallel runs of the original TPA of [4] and outputting a sub-sequence of elements which are $d$ apart.

Let $(b_i)$ be the sequence generated by $k$ parallel copies of the original TPA, thus $\Delta_j = \beta_{j+1} - \beta_j = b_{j+d} - b_j$.

We first show item 1 by bounding $\mathbb{P}\left(z(b_j) - z(b_{j+d}) \geq \varepsilon\right)$, and using

$$\mathbb{P}(b_{j+d} - b_j < \varepsilon) \geq \prod_{i=1}^{d} \mathbb{P}(b_{j+i} - b_{j+i-1} < \varepsilon/d) \,.$$

With the definition of the PPP, and using [3] we have $z(b_i) - z(b_{i+1})$ follows the exponential distribution with mean $1/k$, thus $\mathbb{P}(z(b_i) - z(b_{i+1}) \geq \varepsilon/d) = \exp(-\varepsilon k/d)$. Therefore,

$$\mathbb{P}(z(b_{j+d}) - z(b_j) < \varepsilon) \geq \prod_{i=1}^{d} \mathbb{P}\left(z(b_{j+i}) - z(b_{j+i-1}) < \varepsilon/d\right) = (1 - \exp(-\varepsilon k/d))^d \,.$$

To see item 2 of the Lemma let $z'(\beta)$ be the derivative of $z(\cdot)$ with respect to $\beta$, which is $z'(\beta) = \sum_{x \in \Omega} -H(x) \exp(-\beta H(x))/Z(\beta)$, thus $z'(\beta) \leq 0$. Using the Cauchy–Schwarz inequality we have $z''(\beta) = (\sum_{x \in \Omega} H^2(x) \exp(-\beta H(x)) \sum_{x \in \Omega} \exp(-\beta H(x)) - (\sum_{x \in \Omega} -H(x) \exp(-\beta H(x))^2)/Z^2(\beta) \geq 0$. Therefore,

$$z'(\beta_i) < \frac{z(\beta_{i+1}) - z(\beta_i)}{\beta_{i+1} - \beta_i} < z'(\beta_{i+1}),$$

Thus, $\beta_{i+1} - \beta_i < \frac{z(\beta_i) - z(\beta_{i+1})}{-z'(\beta_i)}$. Note that $-z'(\beta_i) = \mathbb{E}[H(x)], x \sim \pi_{\beta_i}$. Therefore, we have:

$$\mathbb{P}\left(\Delta_i \leq \frac{\epsilon}{\mathbb{E}[H]}\right) \geq \mathbb{P}\left(\frac{z(\beta_i) - z(\beta_{i+1})}{-z'(\beta_{i+1})} \leq \frac{\epsilon}{\mathbb{E}[H]}\right)$$
$$= \mathbb{P}\left(z(\beta_i) - z(\beta_{i+1}) \leq \epsilon\right)$$
$$\geq (1 - \exp(-\epsilon k/d))^d$$

Thus $\mathbb{P}\left(\Delta_i \geq \frac{\epsilon}{\mathbb{E}[H]}\right) \geq 1 - (1 - \exp(-\epsilon k/d))^d \approx d \exp(-\epsilon k/d)$. $\qquad \square$

*Proof of Lemma 2.1.* Note that by Thm 3.1. of [9] we have, $\mathbb{E}[(\bar{f}(\vec{X}_{1:\tau}) - \mathbb{E}(f))^2] \leq \frac{2\tau_{\mathrm{rx}}}{\tau} \mathbb{V}[f]$. Dividing both sides by $\left(\mathbb{E}(f)\right)^2$ we get the second part of the premise. The first part concludes from setting $\tau = \tau_{\mathrm{rx}}$.

$\qquad \square$

## A.3 RELMEANEST

**RELMEANEST in summary** To employ progressive sampling, we start by a small sample size and calculate the *empirical estimation* of the variance at each iteration. We estimate an upper bound on the trace variance based on its empirical estimation, and using that we check a termination condition.

Our variance estimator is what Cousins et al. introduced, and is based on running two independent chains. Each sample is obtained by taking a trace of length $T$ (given upper-bound on relaxation time) and taking the average over all observed values on that trace. Thus, *half the square difference* of the averages on the two chains is an *unbiased estimate* of the *trace variance*.

Before showing the result, we state two key theorems from the literature, which describe how our tail bounds work.

**Theorem A.2** (Hoeffding-Type Bounds for Mixing Processes, (see Thm. 2.1 of [2])). *For any $\delta \in (0, 1)$, we have*

$$\mathbb{P}\left(|\hat{\mu} - \mu| \geq \sqrt{\frac{2(1 + \lambda)(\frac{R^2}{4})\ln(\frac{2}{\delta})}{(1 - \lambda)m}}\right) \leq \delta \,. \tag{2}$$

*This implies sample complexity*

$$m_H(\lambda, R, \varepsilon, \delta) = \frac{1+\lambda}{1-\lambda} \ln(\tfrac{2}{\delta}) \frac{R^2}{2\varepsilon^2} \in \Theta\Big(\tau_{\mathrm{rx}} \ln(\tfrac{1}{\delta}) \frac{R^2}{\varepsilon^2}\Big) \ .$$

**Theorem A.3** (Bernstein-Type Bound for Mixing Process [5, Thm. 1.2]). *For any $\delta \in (0,1)$, we have*

$$\mathbb{P}\left(|\hat{\mu} - \mu| \geq \frac{10R\ln(\tfrac{2}{\delta})}{(1-\lambda)m} + \sqrt{\frac{2(1+\lambda)v_\pi \ln(\tfrac{2}{\delta})}{(1-\lambda)m}}\right) \leq \delta \ . \tag{3}$$

*This implies sample complexity*

$$m_B(\lambda, R, v, \varepsilon, \delta) = \frac{2}{1-\lambda} \ln(\tfrac{2}{\delta})\Big(\frac{5R}{\varepsilon} + \frac{(1+\lambda)v_\pi}{\varepsilon^2}\Big) \in \Theta\Big(\tau_{\mathrm{rx}} \ln(\tfrac{1}{\delta})\Big(\frac{R}{\varepsilon} + \frac{v_\pi}{\varepsilon^2}\Big)\Big) \ .$$

We now show the main result.

*Proof of Theorem 2.2.* Suppose confidence interval $[a, b]$. The interval endpoints, multiplicative error $\varepsilon_\times$, and additive error $\varepsilon_+$ are related as $2\varepsilon_+ = a\frac{1+\varepsilon_\times}{1-\varepsilon_\times} - a = a\frac{2\varepsilon_\times}{1-\varepsilon_\times}$, depicted graphically below.

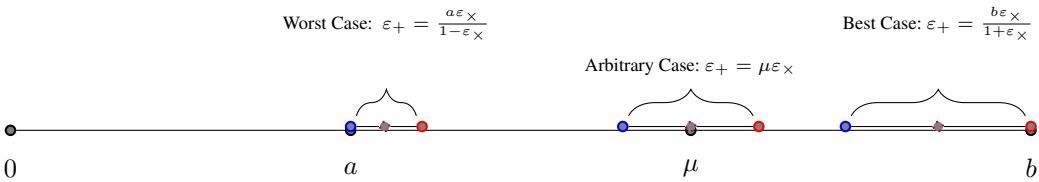

We derive a geometric progressive sampling schedule such that the algorithm draws sample sizes, ranging between optimistic and pessimistic (over unknown variance and mean) upper and lower bounds on the sufficient sample size.

Using the Markov chain Bennett inequality [5], the best-case complexity, assuming maximal expectation, and minimal variance, is

$$\begin{aligned} m^\downarrow &\geq m_B(\Lambda, R, 0, \varepsilon_+, \frac{2\delta}{3I}) \\ &\geq \frac{(1+\Lambda)R\ln\frac{3I}{\delta}}{(1-\Lambda)\varepsilon_+} = \frac{(1+\Lambda)R\ln\frac{3I}{\delta}(1+\varepsilon_\times)}{b(1-\Lambda)\varepsilon_\times} \ . \end{aligned}$$

The worst-case complexity, then assuming minimal expectation, and maximal variance, is

$$\begin{aligned} m^\uparrow &\geq m_H(\Lambda, R, \varepsilon_+, \frac{2\delta}{3I}) \\ &\geq \frac{(1+\Lambda)R^2\ln\frac{3I}{\delta}}{2(1-\Lambda)\varepsilon_+^2} = \frac{(1+\Lambda)R^2\ln\frac{3I}{\delta}(1-\varepsilon_\times)^2}{2(1-\Lambda)a^2\varepsilon_\times^2} \ , \end{aligned}$$

via the Markov chain Hoeffding's inequality [7].

Consequently, a doubling schedule requires $I = \left\lfloor \log_2\left(\frac{m^\uparrow}{m^\downarrow}\right) \right\rfloor = \left\lfloor \log_2\left(\frac{bR}{2a^2} \cdot \frac{(1-\varepsilon_\times)^2}{(1+\varepsilon_\times)\varepsilon_\times}\right) \right\rfloor$ steps.

All tail bounds on variances and means are hold simultanously with probability at least $1 - \delta$ (by union bound), and the doubling schedule never overshoots the sufficient sample size by more than a constant factor, which yields the stated guarantees.

The proof consists of two parts, in both we make derive our new bounds by writing an $\varepsilon_\times$-multiplicative approximation in terms of an $\varepsilon_+$-additive approximation.

In the worst-case, we *underestimate* the true mean $\mu$ by a factor $(1 - \varepsilon_\times)$, and thus require a radius $\varepsilon_+ = \varepsilon_\times(1 - \varepsilon_\times)\mu$ additive confidence interval.

We first show the *correctness guarantee*.

Observe that the sampling schedule is selected such that the final iteration $I$ of the algorithm will draw a sufficiently large sample (size $m^{\uparrow}$) such that the Hoeffding inequality will yield such a confidence interval, even for worst-case (minimal) $\mu$. Now observe that over the course of the algorithm, in each iteration, 3 tail bounds are applied; one to upper-bound the variance, and then two to upper and lower bound the mean in terms of the variance bound) as in [1]. By union bound, all $3I$ tail-bounds hold simultaneously with probability at least $1 - \delta$, thus when the algorithm terminates, it produces a correct answer with at least said probability.

We now show the *efficiency guarantee*. Suppose we get $\hat{\mu}$ from RELMEANEST, by guarantee of correctness of the algorithm, we have a lower bound on $\hat{\mu}$, $\hat{\mu} \geq \mu(1 - \varepsilon_\times)$ with probability at least $1 - \delta$.

Furthermore, we have $\varepsilon_+ = \mu\varepsilon_\times$ and $\mathrm{trv}^{(\tau\mathrm{rx})} = (\mathrm{Reltrv}^{\tau_{\mathrm{rx}}} - 1) \times \hat{\mu}^2 \geq (\mathrm{Reltrv}^{\tau_{\mathrm{rx}}} - 1)\mu^2(1 - \varepsilon_\times)^2$. For this $\varepsilon_+$, we have via the Bernstein inequality that

$$m^* \in \mathcal{O}\left(\log\left(\frac{\log(R/(\mu\varepsilon_\times))}{\delta}\right)\left(\frac{R/\mu}{(1-\Lambda)\varepsilon_\times} + \frac{\tau_{\mathrm{rx}}(\mathrm{Reltrv}^{\tau_{\mathrm{rx}}} - 1)}{\varepsilon_\times^2}\right)\right)$$

would be a sufficient sample size if (1) the algorithm were to draw a sample of this size, and (2) we were to use the *true trace variance* instead of the *estimated upper-bound on trace variance*.

Fortunately, correcting for (1) adds a constant factor to the sample complexity, as the first sample size $\alpha$ is selected to be twice the minimal sufficient sample size $m^{\downarrow}$ (i.e., the sample size such that no smaller sample size would be sufficient), and at each iteration the sample size selected is double the previous (line 8). In other words, this geometric grid will never overshoot any sample size by more than a factor 2.

Resolving (2) is a bit more subtle, but we now show that there is no asymptotic change in replacing the variance with the estimated variance upper bound (w.h.p.). First, note that the Bernstein bound is *bidirectional*, so it can just as well be used to upper-bound empirical variance with true variance as to upper-bound true variance with empirical variance. We bound true variance in terms of empirical variance on line 14, and note that here we have

$$v \leq u \in \hat{v} + \mathcal{O}\left(\frac{R^2 \ln\frac{I}{\delta}}{m} + \sqrt{\frac{R^2 \hat{v} \ln\frac{I}{\delta}}{m}}\right) \quad .$$

Fortunately, the latter terms are negligible, as in line 15, we bound

$$\varepsilon_+ \in \mathcal{O}\left(\frac{R\ln\frac{I}{\delta}}{m} + \sqrt{\frac{u\ln\frac{I}{\delta}}{m}}\right)$$

$$= \mathcal{O}\left(\frac{R\ln\frac{I}{\delta}}{m} + \sqrt{\frac{\left(\hat{v} + \mathcal{O}\left(\frac{R\ln\frac{I}{\delta}}{m} + \sqrt{\frac{\hat{v}\ln\frac{I}{\delta}}{m}}\right)\right)\ln\frac{I}{\delta}}{m}}\right)$$

$$= \mathcal{O}\left(\frac{R\ln\frac{I}{\delta}}{m} + \sqrt{\frac{\left(v + \mathcal{O}\left(\frac{R\ln\frac{I}{\delta}}{m} + \sqrt{\frac{v\ln\frac{I}{\delta}}{m}}\right) + \mathcal{O}\left(\frac{R\ln\frac{I}{\delta}}{m} + \sqrt{\frac{\hat{v}\ln\frac{I}{\delta}}{m}}\right)\right)\ln\frac{I}{\delta}}{m}}\right) \quad \text{(w.h.p.)}$$

$$= \mathcal{O}\left(\frac{R\ln\frac{I}{\delta}}{m} + \sqrt{\frac{v\ln\frac{I}{\delta}}{m}}\right) \quad . \quad \text{(w.h.p.)}$$

Putting these together, we thus have that, w.h.p., sample consumption is bounded as

$$\hat{m} \in 2\mathcal{O}(m^*) = \mathcal{O}\left(\log\left(\frac{\log(R/(\mu\varepsilon_\times))}{\delta}\right)\left(\frac{R/\mu}{(1-\Lambda)\varepsilon_\times} + \frac{\tau_{\mathrm{rx}}(\mathrm{Reltrv}^{\tau_{\mathrm{rx}}} - 1)}{\varepsilon_\times^2}\right)\right) \quad.$$

To conclude, we need only relate $T(\mathrm{Reltrv}^T - 1)$ and $\tau_{\mathrm{rx}}(\mathrm{Reltrv}^{\tau_{\mathrm{rx}}} - 1)$. Letting $T$ as in line **??**, note that since $T \geq \tau_{\mathrm{rx}}$, it holds that $T(\mathrm{Reltrv}^T - 1) \geq \tau_{\mathrm{rx}}(\mathrm{Reltrv}^{\tau_{\mathrm{rx}}} - 1)$, by the trace variance inequalities, which yields the result. □

### A.4 Missing proofs from analysis of SUPERCHAINTRACEGIBBS

*Proof of thm 2.4.* Follows immediately from thm. 2.2 and plugging in the values for paired product estimators and the product chain. □

*Full Proof of Lemma 2.8.* Let $\bar{\beta}_{i,i+1} \doteq \frac{\beta_i + \beta_{i+1}}{2}$, we have $\mu_i = \frac{Z(\bar{\beta}_{i,i+1})}{Z(\beta_i)}$ and $\nu_i = \frac{Z(\bar{\beta}_{i,i+1})}{Z(\beta_{i+1})}$. Thus we have $\nu = \frac{\prod_{i=1}^{\ell-1} Z(\bar{\beta}_{i,i+1})}{\prod_{i=1}^{\ell-1} Z(\beta_{i+1})} > 1$, $\mu = \frac{\prod_{i=1}^{\ell-1} Z(\bar{\beta}_{i,i+1})}{\prod_{i=1}^{\ell-1} Z(\beta_i)} < 1$.

Note that $\nu = \mu \frac{Z(\beta_0)}{Z(\beta_{\max})}$, thus we proceed by bounding $\mu$.

$$\log \prod_{i=1}^{\ell-1} Z(\bar{\beta}_{i,i+1}) = \sum_{i=1}^{\ell-1} z(\bar{\beta}_{i,i+1}) \qquad \text{TAKING log}$$

$$\geq \sum_{i=1}^{\ell-1} z(\beta_i) - \frac{\Delta_i}{2} \mathbb{E}_{x \sim \pi_{\beta_i}}[H(x)] \quad \text{TAYLOR EXPANSION \& THAT } \frac{\partial^2}{\partial\beta^2} z(\beta) > 0$$

Thus, by taking exponents we get:

$$\prod_{i=1}^{\ell-1} Z(\bar{\beta}_{i,i+1}) \geq \exp\left(\sum_{i=1}^{\ell-1} z(\beta_i) - \frac{\Delta_i}{2} \mathbb{E}_{x \sim \pi_{\beta_{i,i+1}}}[H(x)]\right)$$

$$\geq \left(\prod_{i=1}^{\ell-1} Z(\beta_i)\right) \exp\left(-\sum_{i=1}^{\ell-1} \frac{\Delta_i}{2} \mathbb{E}_{x \sim \pi_{\beta_i}}[H(x)]\right)$$

Therefore, $\mu = \frac{\prod_{i=1}^{\ell-1} Z(\bar{\beta}_{i,i+1})}{\prod_{i=1}^{\ell-1} Z(\beta_i)} \geq \exp\left(-\sum_{i=1}^{\ell-1} \frac{\Delta_i}{2} \mathbb{E}_{x \sim \pi_{\beta_i}}[H(x)]\right)$. Using this form, we now employ the fundamental theorem of calculus to prove the premise:

Let $\Delta_{\max} \doteq \max_i \Delta_i$.

$$\mu \geq \exp\left(-\sum_{i=1}^{\ell-1} \frac{\Delta_i}{2} \mathop{\mathbb{E}}_{x \sim \pi_{\beta_i}}[H(x)]\right)$$

$$= \exp\left(-\sum_{i=1}^{\ell-1} \frac{\Delta_i}{2} \mathop{\mathbb{E}}_{x \sim \pi_{\beta_i}}[H(x)]\right)$$

$$\geq \exp\left(\frac{1}{2}\int_{\beta_{\min}-\Delta_{\max}}^{\beta_{\max}-\Delta_{\max}} -\mathop{\mathbb{E}}_{x \sim \pi_\beta}[H(x)]\,\mathrm{d}\beta\right) \qquad\qquad \textsc{Increasing Integrand}$$

$$= \exp\left(\frac{1}{2}\big(z(\beta_{\max}-\Delta_{\max}) - z(\beta_{\min}-\Delta_{\max})\big)\right) \qquad \textsc{FTOC and that } z'(\beta) = \mathbb{E}_{x \sim \pi_\beta}H$$

$$\geq \exp\left(\frac{1}{2}\big(z(\beta_{\max}) - z(\beta_{\min}-\Delta_{\max})\big)\right) \qquad\qquad\qquad z \textsc{ is Decreasing}$$

$$= \exp\left(\frac{1}{2}\big(z(\beta_{\max}) - z(\beta_{\min}) + z(\beta_{\min}) - z(\beta_{\min}-\Delta_{\max})\big)\right)$$

$$\geq Q^{-\frac{1}{2}}\sqrt{\frac{Z(\beta_{\min})}{Z(\beta_{\min}-\Delta_{\max})}}\,.$$

From the above we also conclude that $\nu \geq Q^{1/2}\sqrt{\frac{Z(\beta_{\min})}{Z(\beta_{\min}-\Delta_{\max})}}$. Note that $\mathrm{Range}(f) = \exp(-\frac{\Delta}{2}H_{\min}) - \exp(-\frac{\Delta}{2}H_{\max}) \leq \sqrt{\exp(-\Delta H_{\min})}$ and $\mathrm{Range}(g) = \exp(\frac{\Delta}{2}H_{\max}) - \exp(\frac{\Delta}{2}H_{\min}) \leq \sqrt{\exp(\Delta H_{\max})}$. Thus the lemma is concluded.

$\square$

*Proof of Corollary 2.6.* The corollary follows from thm 2.2 plugging in $R$ from lemma 2.5 and setting $\tau_{\mathrm{prx}} = \ell \max_{i=1}^{\ell} \tau_i$ (see, e.g., [8]). $\square$

## A.5   Analysis of PARALLELTRACEGIBBS

Let $(\beta_0, \beta_1, \dots \beta_l)$ be a cooling schedule generated by $\textsc{Tpa}(k, d)$, where $k$ and $d$ are chosen as in [6]. For each $i$ let $f_{\beta_i,\beta_{i+1}}, g_{\beta_{i-1},\beta_i}$ be the paired estimators corresponding to this schedule, and $\mu_i = \mathbb{E}[f_{\beta_i,\beta_{i+1}}], \nu_i = \mathbb{E}[g_{\beta_{i-1},\beta_i}]$. PARALLELTRACEGIBBS estimates $Q$ by running RELMEANESTon each $\mathcal{G}_{H,\beta_i}$, to estimate $\mu_i$ and $\nu_i$s each with precision $\varepsilon' = (\sqrt[l]{1+\varepsilon}-1)/(\sqrt[l]{1+\varepsilon}+1)$. Note that by this setting, $Q$ will be approximated within multiplicative factor of $\left(1+\varepsilon'/1-\varepsilon'\right)^\ell$. Assume $\tau_i$ is the true relaxation time of $\mathcal{G}_{H,\beta_i}$ and suppose $\Lambda_i$ is a known upper bound on the second eigenvalue of $\mathcal{G}_{H,\beta_i}$, thus $(\Lambda_i - 1)^{-1}\log(2) \geq \tau_i$. The following hold and thm 2.7 is immediately concluded from it:

**Lemma A.4.** *Let $H_{\max} \doteq \max_{x \in \Omega} H(x)$. we have:*

*1. for all $1 \leq i \leq \ell$, $\mathrm{Range}(f_{\beta_i,\beta_{i+1}})/\mu_i \leq \ell^{1/\log(n)}$,*

*2. for all $1 \leq i \leq \ell$, $\mathrm{Range}(g_{\beta_{i-1},\beta_i})/\nu_i \leq \ell^{\alpha_0(i)/\log n}$, where $\alpha_0(i) = \left(H_{\max}/2\mathbb{E}[H(x)]\right) - 1$, $x \sim \pi_{\beta_i}$.*

*Proof.* Let $\Delta_i = \beta_{i+1} - \beta_i$. Thus, $f_i(x) = \exp\left(\frac{-\Delta_i}{2}H(x)\right)$ and $g_i(x) = \exp\left(\frac{\Delta_i}{2}H(x)\right)$. So we have:

$$\mathrm{Range}(f_i) = \exp\left(\frac{-\Delta_i}{2}\min_x H(x)\right) - \exp\left(\frac{-\Delta_i}{2}\max_x H(x)\right) \leq \exp\left(\frac{-\Delta_i}{2}\min_x H(x)\right)$$

and

$$\mathrm{Range}(g_i) = \exp\left(\frac{\Delta_i}{2}\max_x H(x)\right) - \exp\left(\frac{\Delta_i}{2}\min_x H(x)\right) \leq \exp\left(\frac{\Delta_i}{2}\max_x H(x)\right)$$

$$\mu_i = Z(\beta_i + \Delta_i/2)/Z(\beta_i) \quad \& \quad \nu_i = Z(\beta_{i+1} - \Delta_i/2)/Z(\beta_{i+1})$$

$$\frac{\text{Range}(f_i)}{\mu_i} \leq \frac{\exp\left(\frac{-\Delta_i}{2}\min_x H(x)\right)}{\exp\left(z(\beta_i + \Delta_i/2) - z(\beta_i)\right)}, \frac{\text{Range}(g_i)}{\nu_i} \leq \frac{\exp\left(\frac{\Delta_i}{2}\max_x H(x)\right)}{\exp\left(z(\beta_{i+1} - \Delta_i/2) - z(\beta_{i+1})\right)}$$

(4)

Writing $\Delta_i/2 = \frac{\Delta_i/2}{z(\beta_i + \Delta_i/2) - z(\beta_i)}(z(\beta_i + \Delta_i/2) - z(\beta_i))$, we get:

$$\frac{\text{Range}(f_i)}{\mu_i} \leq \exp\left(-\frac{\Delta_i}{2}\min_x H(x) - \left(z(\beta_i + \Delta_i/2) - z(\beta_i)\right)\right)$$

$$\leq \exp\left(\left(z(\beta_i + \Delta_i/2) - z(\beta_i)\right)\left(\frac{-\Delta_i \cdot \min_x H(x)}{2\left(z(\beta_i + \Delta_i/2) - z(\beta_i)\right)} - 1\right)\right)$$

and

$$\frac{\text{Range}(g_i)}{\nu_i} \leq \exp\left(\left(z(\beta_{i+1} - \Delta_i/2) - z(\beta_{i+1})\right)\left(\frac{\Delta_i \cdot \max_x H(x)}{2\left(z(\beta_{i+1} - \Delta_i/2) - z(\beta_{i+1})\right)} - 1\right)\right)$$

Let $z'$ and $z''$ be the first and second derivative of $z$ with respect to $\beta$. Note that $z'(\beta) = \mathbb{E}_{x\sim\pi_\beta}[-H(x)]$. Since $z'' \geq 0$ we have:

$$z'(\beta_i) < \frac{z(\beta_i + \Delta_i/2) - z(\beta_i)}{\Delta_i/2} < z'(\beta_i + \Delta_i/2)$$

and

$$z'(\beta_{i+1} - \Delta_i/2) < \frac{z(\beta_{i+1}) - z(\beta_{i+1} - \Delta_i/2)}{\Delta_i/2} < z'(\beta_{i+1}).$$

Which are equivalent to $\frac{1}{z'(\beta_i + \Delta_i/2)} \leq \frac{\Delta_i/2}{z(\beta_i + \Delta_i/2) - z(\beta_i)} \leq \frac{1}{z'(\beta_i)}$ and $\frac{1}{z'(\beta_{i+1})} \leq \frac{\Delta_i/2}{z(\beta_{i+1}) - z(\beta_{i+1} - \Delta_i/2)} \leq \frac{1}{z'(\beta_{i+1} - \Delta_i/2)}$.

Therefore,

$$\frac{\text{Range}(f_i)}{\mu_i} \leq \exp\left(\left(z(\beta_i + \Delta_i/2) - z(\beta_i)\right)\left(\frac{-\min_x H(x)}{2}\frac{1}{z'(\beta_i)} - 1\right)\right) \tag{5}$$

$$= \exp\left(\left(z(\beta_i + \Delta_i/2) - z(\beta_i)\right)\left(\frac{\min_x H(x)}{2}\frac{1}{\mathbb{E}[H]} - 1\right)\right) \tag{6}$$

$$\leq \exp\left(z(\beta_i) - z(\beta_i + \Delta_i/2)\right) \tag{7}$$

Similarly for range of $g_i$s we have:

$$\frac{\text{Range}(g_i)}{\nu_i} \leq \exp\left(\left(z(\beta_{i+1} - \Delta_i/2) - z(\beta_{i+1})\right)\left(\frac{-\max_x H(x)}{2}\frac{1}{z'(\beta_i)} - 1\right)\right) \tag{8}$$

$$\leq \exp\left(\left(z(\beta_{i+1} - \Delta_i/2) - z(\beta_{i+1})\right)\left(\frac{\max_x H(x)}{2\mathbb{E}_{\pi_{\beta_i}}[H(X)]} - 1\right)\right) \tag{9}$$

We now use (5) together with lemma A.1. Setting $d = 1$ we have,

$$\mathbb{P}\left(z(\beta_i) - z(\beta_{i+1}) > \frac{\log(3l/4)}{\log n}\right) = \exp(-\frac{\log(3l/4)}{\log n} \cdot k) = (3/4)\exp(-\log l / \log n(\log n))$$

$$= (3/4)(1/l).$$

Using union bound over all $1 \leq i \leq \ell$ and that $z(\beta_i) - z(\beta_{i+1}) \geq z(\beta_i) - z(\beta_i + \Delta_i/2)$, we conclude that with probability at least $3/4$ we have that for all $f_i$, $\text{Range}(f_i)/\mu_i \leq \ell^{1/\log(n)}$.

Similarly using (8), the union bound, lemma A.1 and that $z(\beta_i) - z(\beta_{i+1}) \geq z(\beta_i - \Delta_i/2) - z(\beta_{i+1})$, we can show that with constant probability all $g_i$s generated by the TPA schedule obey: $\forall g_i; 1 \leq i \leq \ell$, $\text{Range}(g_i)/\nu_i \leq \exp\left((\log l/\log n) \cdot (\alpha)\right) = \ell^{\alpha_0/\log n}$, where $\alpha_0 = \frac{\max_x H(x)}{2\mathbb{E}_{\pi_{\beta_i}}[H(X)]} - 1$. $\qquad\square$

The following corollary is concluded from lemma A.4 and relative trace variance bounds:

**Corollary A.5.** *When* $\varepsilon \leq \ell^{1/\log(n)}(1 + \ell^{\alpha_0(i)}) \cdot \frac{\ell\tau_{\beta_i}}{(1-\Lambda_i)^{-1}}$, RELMEANEST *invoked on the ith iteration will stop using sample consumption of* $\tilde{O}\left(\ell^2 \tau_i \text{Reltrv}_i\right)$ *note that this is improvement over classic bounds which are* $\tilde{O}\left((1-\Lambda_i)^{-1}\mathbb{V}\text{rel}_i\right)$. *In total the sample complexity of* PARALLELTRACEGIBBS *for* $\varepsilon \leq \ell^{1/\log(n)} \min_i(1 + \ell^{\alpha_0(i)}) \cdot \frac{\ell\tau_{\beta_i}}{(1-\Lambda_i)^{-1}}$ *is dominated by* $\tilde{O}\left(\ell^2 \sum_{i=1}^{\ell} \tau_i \text{Reltrv}_i\right)$ .

## A.6 Further experimental results

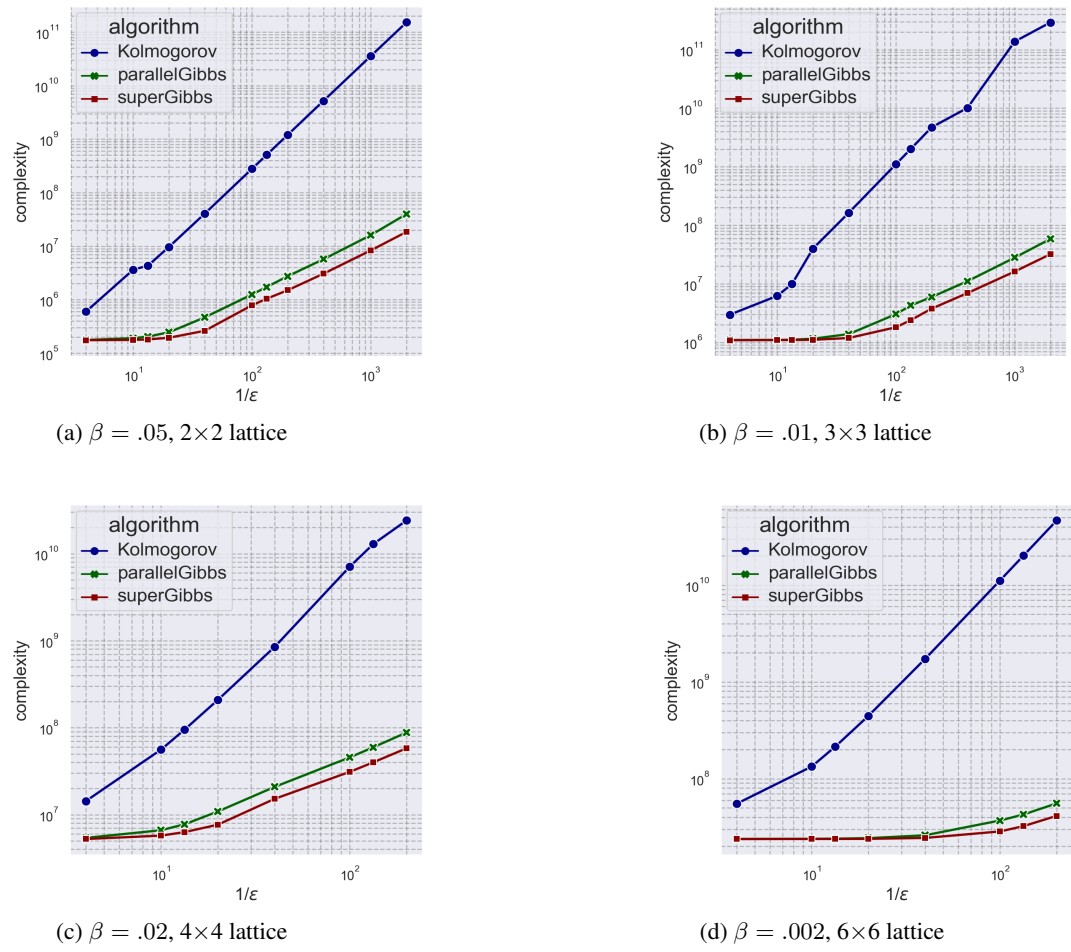

(a) $\beta = .05$, 2×2 lattice

(b) $\beta = .01$, 3×3 lattice

(c) $\beta = .02$, 4×4 lattice

(d) $\beta = .002$, 6×6 lattice

Figure 1: Comparison of sample complexity on Ising models.