# OpenReview forum: "Fast Doubly-Adaptive MCMC to Estimate the Gibbs Partition Function with Weak Mixing Time Bounds"
_NeurIPS.cc/2021/Conference — NeurIPS 2021 Poster_

### Official Review · Reviewer_AZX6 · 2021-06-27

**Rating:** 7
**Confidence:** 3

**Summary:**

In this paper, the authors introduce a new method to estimate the partition functions of Gibbs distributions. They try to improve the classical mean estimate computation by using an adaptive MCMC mean estimator. And they justify the priority of their methods through theoretical analysis and experiments.

**Limitations And Societal Impact:**

Some limitations of this paper are listed as follows: (The reviewer is not an expert in this area and might misunderstand something.)
1. In the contribution section (Sec 1.2), the reviewer thinks the authors can delete lines 121-129 since all four points explain how the paper achieves an improvement in computational complexity.
It's good that the authors explain how they obtain the improvement. But they are actually not the main contributions, the authors should move them somewhere else.
2. In the conclusion section (Sec 4), Table 1 compares the computation complexity of different methods. But the authors omit the $1/\epsilon$ term. It's true that this term can be omitted when $\epsilon$ is small enough, but the authors should mention the conditions that this holds. The authors also mention $RelR_i$ is a large number, then it seems we should choose $\epsilon$ super small to see the priority of new methods.
3. According to Table 1, the authors don't give any comparison between $\text{Reltrv}^{\tau_i}$ and $V_{rel}$. If they are in the same order, the reviewers think the new method is better because $\ln(l/\epsilon)>l^2$ when $\epsilon$ is small. Then it seems that the main improvement comes from the new bound eliminates the $\ln(1/\epsilon)$ factor. This seems a small improvement. Can the authors give some explanation why this is important?

As a potential reader, the reviewer thinks that there are some confusing places in the paper:
1. In line 59, "the values of random terms in the sum have large variance". According to (1), the reviewer didn't see any randomness. Does the author mean the variance of $\pi_{\beta}$ is large so $Z(\beta)$ is hard to estimate?
2. In line 83, $"O(log(Q)log(H_{\max})"$, this is the first time $H_{\max}$ appears. But the authors define it in Lemma 2.6 line 231. The authors can move the definition forward or at least give an explanation in line 83.
3. In line 84, "TPA(k,d)", this is the first time $d$ appears. Again, there is no definition and explanation.
4. In lines 100 and 108, what is $\pi_\min$? The reviewer didn't find its definition in the paper.
5. In the results of Section 2.2, what is $\epsilon$?
6. In line 303, according to Table 1, the reviewers think "$\ln(l)$" should be "$\ln(l/\epsilon)$".
7. In line 305 Table 1, what is $T_i$? Is it the same as $\tau_i$?
8. As said before, the comparison in Table 1 is very unclear. The authors should spend more space to do the comparison and make it clear.


**Main Review:**

Overall, the reviewer thinks this is a good paper. As claimed by the authors, they introduce a new method that is proven to outperform the state of art algorithms. And the experiments also justify this.

However, the reviewer thinks this paper is not very clear. The reviewer is not familiar with this problem and feels it's a little hard to follow this paper. The reviewer will point out some confusing parts in "Limitations".

**Time Spent Reviewing:**

2 hours

---

> ### Author Response · Authors · 2021-08-10
> **Response to reviewer AZX6**
>
> We thank the reviewer for carefully reviewing our paper and providing  detailed comments. Based on these comments, we will  make our exposition more clear in the extra page available in the camera ready version.
>
> In particular, we will add to section 2 that $\epsilon$ is a parameter given as input to the algorithm and we guarantee a $(1+\epsilon)$ approximation to the partition function.
>
> We will add an explanation about parameter $d$ in the TPA method, and that Kolmogorov [K] modified Huber's original algorithm, and added this parameter to the algorithm. In the current version, it is only stated as a footnote that $d=64$ and we have presented both Huber [H] and Kolmogorov's [K] version of the TPA algorithm in Appendix A.1. We will move this definitions to more suitable locations in the text.
>
> We will also add the missing definitions like $\pi_{\min}\doteq \min_{x\in \Omega} \pi(x)$.
>
> Furthermore, in the additional space we have, we will clarify Table 1, and add the following explanations:
> for each $i$, $\tau_i$ is the relaxation time of the chain at temperature $\beta_i$ and
> $T_i$ is a known upper bound on it. Thus, in general $T_i\geq \tau_i$ and often $T_i\gg \tau_i$. One of the main advantages over Kolmogorov's is that of the complexity of our algorithm  for small $\epsilon$  is dependent on true relaxation time $\tau_i$ rather than loose upper bounds $T_i$.
> We thank the reviewer for pointing out that adding $1/\epsilon$ to the table will help the table's comprehension, and we will fix this issue.
>
>
>
> [H]  Huber, Mark. "Approximation algorithms for the normalizing constant of Gibbs distributions." The Annals of Applied Probability 25.2 (2015): 974-985.
>
>
>
> [K] Kolmogorov, Vladimir. "A faster approximation algorithm for the Gibbs partition function." COLT 2018.

---

> > ### Comment · Reviewer_AZX6 · 2021-08-11
> > **Response to authors**
> >
> > The reviewer is satisfied with the authors' response. The reviewer thinks the authors have addressed all the concerns.
> >
> > The reviewer is not very familiar with this area. So the reviewer prefers to leave the discussion to others.

---

### Official Review · Reviewer_TAYx · 2021-07-12

**Rating:** 4
**Confidence:** 5

**Summary:**

The objective of this work is to define a procedure for calculating Gibbs distribution normalization constants. The calculation of normalisation constants is a central problem in many fields and most known methods depend on many hyper-parameters that are difficult to optimise. The most efficient methods (such as "paired product estimator" extending "multiple  sampling" of Valleau and Card) are based on annealing schemes. The normalisation constant is obtained in this case as a product, each term of the product being calculated by means of an MCMC algorithms. Several strategies have been proposed to optimise the annealing schedule. The optimisation of the annealing schedule requires a measure of the complexity of generating samples with the MCMC algorithm, which typically involves using upper bounds on their mixing times. Despite progress in this direction, the estimation of chain mixing times is a difficult subject: the known bounds are, with few exceptions (of very simple models with a lot of symetries), very pessimistic.
The paper presents an imporoved mean estimator for comuting the nomalizing constant of a Gibbs distribution. The proposed algorithm is claimed to improve SOTA results in this field. Compared to other techniques, the proposed algorithm is "less dependent" (the authors words) on the tightness of the bounds use to measure the mixing time. The main improvements bought by the authors is clained to be high-accuracy computaitons.

**Main Review:**


Major comments:
Except for the presentation of the basic algorithm (which is fairly classical), the paper is difficult to read. It is difficult to understand the contributions, the results, I was  quite confuse even after reading the full paper. The discussion and the proofs use a lot of external references, which require each time to understand other conventions, notations, concepts. It would be much better to recap the required results in a Section in the supplement paper, to ease the reading.

I have tried to check the proof of Theorem 2.4 in the supplementary version for example, and I found it quite difficut to follow. I would definitely prefer in the supplement paper to get  complete and detailed arguments, not a sketch of proofs with external references (when you cite an external result in a paper is definitely better to get a precise reference to the theorem or proposition which is used, it is very complicated otherwise).

I am surprised not to see in the complexity bounds the initial distribution of the Markov chain. The authors use the results  of Jiang, Sun, Fan (2018) [Theorem 1 and 2] which both assumed that the MC is stationary. It would be interesting to know how the authors plan to work with the non-stationary case.


The assumption of knowing an upper bound of the spectral gap seems difficult to me. The estimators are, except for examples, generally extremely pessimistic, especially for general state chains.
Minor comments
- I find the discussion at the beginning of Section 2 difficult to follow
- Lemma 1 is dificult to understand because some quantities which are presented are not defined (what is  $\Delta_i$ ? )
- line 177: "estimating means.. union bound)." difficult to understand as it is stated.

**Time Spent Reviewing:**

3

---

> ### Author Response · Authors · 2021-08-10
> **Response to Reviewer TAYx**
>
> (1) We agree with the reviewer's critique that understanding of paper relies on understanding the techniques in [H] and [CHU]. We tried to present briefly the techniques used from these resources  in 9 pages. Furthermore, in section A.1. of appendix we explain the TPA schedule, which is taken from [H]. From [CHU] we use their variance estimator,
>  we will add a  summary of the estimator's functionality and guarantees   in the appendix. The concept of the $\textit{relative trace variance}$ that we introduce here is related to $\textit{trace variance}$ of [CHU],  however here it stands on its own and is fully explained in the main body of the paper.
> We believe section A.3. and proof of Theorem 2.4 stand on their own.
>
>
> (2) For handing non-stationarity, in our experiments we have applied the standard  warm-start technique, i.e., we initially ran each chain for $T_0=\tau_{\rm rx}(\ln \pi_{\min})^{-1}$ (a.k.a. the uniform mixing time)  steps. Note that since warm start is only performed once, it  only effects  our bounds slightly  (see any MCMC textbook e.g., [LPW] or Chapter A.3.3. of [CHU]). We thank the reviewer for pointing this out and we will add this discussion into to the paper.
>
>
>
>
> (3)
> Upper bounds on the spectral gaps of Gibbs samplers are widely available in the literature (references are provided in the introduction and related work).
> Our work  builds  on a series of work whose aim is to provide rigorous mathematical analysis and they all assume  upper bounds on spectral gap.
> We improve these results in terms of dependency on spectral gap bounds and show  our sample complexity depends on true relaxation time and not its loose upper-bounds for high precision estimation (see theorem 2.4).
>
> (4) The significance of our work is to provide an algorithm with rigorous mathematical proofs and guarantees which always hold. Our work diverges from heuristics which may perform poorly on sophisticated Gibbs samplers. Our goal is to be able to handle all, including pessimistic, cases.
>
> $\textbf{Response to minor comments}$
>
> (5)  $\Delta_i$ are defined as $\Delta_i\doteq\beta_{i+1}-\beta_i$. This is written in line 525 of the proofs section. Unfortunately we forgot to move it to the main statement. We thank the reviewer and will fix this issue in the paper.
>
>
> (6) The unclear part in line 177 is referring to [H]'s work and it is not our paper's contribution.
> In our work we have introduced \emph{the relative trace variance} which is extension of the \emph{relative variance}  (applicable to  i.i.d.\ samples) used by [H].
>  In these paragraphs, we are explaining [H]'s work and how by bounding the relative variance (not variance) of $F\doteq \prod_{i=1}^lf_{\beta_i\beta_{i+1}}$ tightly, they manage to prove smaller sampler complexity  compared to an alternative approach of  bounding variances (not relative variances) of all  $f_{\beta_i\beta_{i+1}}$ separately, and estimating their means separately  with higher precision (because of the need for a union bound over all $i=1,2,\dots , l$).
>
>
>  [SRO]
> De Sa, Kunle Olukotun, Christopher Ré
>  ``Ensuring Rapid Mixing and Low Bias for Asynchronous Gibbs Sampling '', ICML 2016.
>
>
>
>  [DZOR] De Sa, C., Zhang, C., Olukotun, K., & Ré, C. ``Rapidly mixing gibbs sampling for a class of factor graphs using hierarchy width'' (Neurips 2015).
>
>  [JLP] Ding, Jian, Eyal Lubetzky, and Yuval Peres. ``The mixing time evolution of Glauber dynamics for the mean-field Ising model.'' Communications in Mathematical Physics  (2009).
>
>
> [H]  Huber, Mark. "Approximation algorithms for the normalizing constant of Gibbs distributions." The Annals of Applied Probability 25.2 (2015): 974-985.
>
> [CHU] Cyrus Cousins, Shahrzad Haddadan, Eli Upfal, ``Making mean-estimation more efficient using an MCMC trace variance approach: DynaMITE'' ArXiv.
>
> [LPW] Levin, David A., and Yuval Peres. Markov chains and mixing times. Vol. 107. American Mathematical Soc., 2017.

---

> > ### Comment · Reviewer_TAYx · 2021-08-31
> > **Response to the authors**
> >
> > The authors' responses are clear and do not fundamentally change my assessment of the paper. I do not dispute that it's a solid paper, but I find that it's basically pretty old ideas, improvements that seem incremental and justify my grade. I would have a different opinion if the paper had been flawless in form, but that is not the case in its current form (it undoubtedly will be in a revision)

---

### Official Review · Reviewer_YZ7S · 2021-07-16

**Rating:** 6
**Confidence:** 3

**Summary:**

The authors propose a method for estimating the partition function of Gibbs distributions. The method primarily relies on an adaptive mean estimator, and is shown to have theoretical and practical advantages over previous methods.

**Limitations And Societal Impact:**

Yes.

**Main Review:**

The problem of approximating the partition function of discrete probabilistic models is relevant and has a lot of ML applications. This paper mainly focuses on improving previous methods based on paired product estimators [1] by introducing an improved mean estimator that does not require black box (approximately) independent samples. Based on previous work on trace averaging [2], they introduce an adaptive mean estimator, and theoretically show that the resulting method is superior to the state of the art. The final algorithms (Alg. 2 in the paper) and their upper bounds are an interesting contribution, albeit not completely novel, since both adaptive parts (cooling and mean estimation) are largely based on previous work.

I see two major downsides with this paper:
- The exposition of the main theoretical results in section 2 suffers from lack of clarity. Both sections 2.1 and 2.2 are heavy on notation, and contain very little intuitive explanation.
- The experimental section only discusses very small toy examples, which makes me wonder whether the proposed algorithms would actually be applicable to any practically relevant ML problem.

&nbsp;
### References
[1] Mark Huber and Sarah Schott. Random construction of interpolating sets for high-dimensional integration. Journal of Applied Probability, 2014.\
[2] Cyrus Cousins, Shahrzad Haddadan, and Eli Upfal. Making mean-estimation more efficient using an MCMC trace variance approach: DynaMITE. ArXiv, 2020.

**Time Spent Reviewing:**

4

---

> ### Author Response · Authors · 2021-08-10
> **Response to Reviewer YZ7S**
>
> We thank the reviewer for their comments, and will add more intuition to chapter 2 using the extra page available in the camera ready version. To answer the reviewer's critique about our experiments, we would like to emphasize that the goal of our paper is to develop an algorithm with mathematically sound performance guarantees  and proven finite sample   complexities (and not heuristics). Our experiments follow the settings in related theoretical  papers; see, e.g., [PSBR] who run their experiments on graphs having 12 and 15  vertices, and [HS] who only run their experiments on a $4\times 4$ grid.
> Note that our experiments are done on an Ising model on $6\times 6$ grid ($36$ vertices) and for voting models on with $7$ and $11$ vertices ($n=3,5$).
>
> We feel that rigorous guarantees and mathematical analysis for MCMC settings are important as they are increasingly employed in real world machine learning applications, where safety, ethical, and legal concerns preclude the use of algorithms with only heuristic guarantees.
>
> [PSBR] Adarsh Prasad, Vishwak Srinivasan, Sivaraman Balakrishnan, Pradeep Ravikumar,
> ``On Learning Ising Models under Huber's Contamination Model''
> Advances in Neural Information Processing Systems 33 (NeurIPS 2020).
>
>
> [HS] Huber, Mark, and Sarah Schott. ``Using TPA for Bayesian inference.'' Bayesian Statistics 9 (2010): 257-282

---

### Official Review · Reviewer_DZFN · 2021-07-19

**Rating:** 5
**Confidence:** 4

**Summary:**

The authors propose a doubly "adaptive" MCMC algorithm to approximate the partition function in gibbs distributions.
In related work, the partition function of Gibbs distributions are computed using paired product estimators [1].
Such estimators use a cooling schedule $[\beta_{(\cdot)}]$ to estimate two product functions $F$ and $G$ using stationary samples from the Gibbs distribution at each temperature.
The ratio of these two products is then used to compute the partition function.

In the current work, the first "adaptive" refers to the adaptive TPA cooling schedule to decide $[\beta_{(\cdot)}]$.
The second refers to the automatic number of steps used in the MCMC MeanEstimator which is used to estimate the two products mentioned above $F$ and $G$.

[1] - Huber, Mark. "Approximation algorithms for the normalizing constant of Gibbs distributions." The Annals of Applied Probability 25.2 (2015): 974-985.

**Limitations And Societal Impact:**

Adequately addressed

**Main Review:**

### Strengths

- The method is well motivated and the theoretical analysis is very thorough.

### Weaknesses

- The usage of the word adaptive (as an auto-decision on the number of steps) is highly misleading. An entire line of work [2] exists where the word adaptive refers to changing the transition kernel to improve the efficiency of the MCMC.
- In line 77 describing Huber and Schott's estimator, [1] makes statements on estimating $z=\log(Z)$. Please refer to the discussion 2 paragraphs before algorithm 2.2 in [1] ([29] in reviewed paper). (Maybe I am missing something but in order to apply linearity of expectations to a product and to compute an unbiased estimator of a quotient, the log domain makes more sense. However please correct me if I am missing something.)
- Please consider comparison with Annealed Importance Sampling (AIS) [3] which samples a single transition at each temperature such that stationary samples at each temperature aren't needed.
  - I may have missed a nuance which prohibits the use of AIS, and if that is the case I am willing to change my rating.
  - If AIS can indeed be used in this scenario, the second "adaptation" can only be justified thorough empirical evaluation.
- Please add error bars to all plots not just Fig 1.e.
- Please use larger font in plots.

### Other criteria

- Originality : Existing problem, novel solution which combines [1,4] using original tools.
- Quality : Theory is very rigorous, empirical evaluation lacking.
- Clarity : Motivations are well explained, plots need work and experimental evaluation is lacking.
- Significance : Unclear, refer to comment regarding AIS in Weaknesses.

### Note

I am willing to update my review, especially my overall score, based on a strong rebuttal arguing that either AIS is not applicable in this task or is demonstrably bad when empirically evaluated against the method.

[1] - Huber, Mark. "Approximation algorithms for the normalizing constant of Gibbs distributions." The Annals of Applied Probability 25.2 (2015): 974-985.

[2] - Roberts, Gareth O., and Jeffrey S. Rosenthal. "Examples of adaptive MCMC." Journal of computational and graphical statistics 18.2 (2009): 349-367.

[3] - Neal, Radford M. "Annealed importance sampling." Statistics and computing 11.2 (2001): 125-139.

[4] - Cousins, Cyrus, Shahrzad Haddadan, and Eli Upfal. "Making mean-estimation more efficient using an MCMC trace variance approach: DynaMITE." arXiv preprint arXiv:2011.11129 (2020).

**Time Spent Reviewing:**

4

---

> ### Author Response · Authors · 2021-08-10
> **Response to Reviewer  DZFN**
>
> We thank the reviewer
> for their valuable comments. We will add error bars to our figures and clarify that the word "adaptive'' refers to the way Kolmogorov and Huber [H,K] use the term "adaptive schedule'' we will change our paper's title and make clear the distinction with [RR]'s work.
> We now respond to some of the reviewer's questions:
>
> $\textbf{Question.}$ "Maybe I am missing something but in order to apply linearity of expectations to a product and to compute an unbiased estimator of a quotient, the log domain makes more sense. However please correct me if I am missing something''
>
> $\textbf{ Answer.}$
> The reviewer is correct that $z=\log Z$ is analysed for estimating $Z$ in work of [H] (because $z$ will form a Poisson point process). However, the samples taken from each temperature are independent, and rather than linearity of expectation, they simply use $\mathbb{E}[\prod_{i=1}^l X_i]=\prod_{i=1}^l \mathbb{E}[ X_i]$, which holds for independent $X_i$s, to bound $V_{\rm rel}[F]+1$ (and similarly $V_{\rm rel}[G]+1$). (Note that $V_{\rm rel}[F]+1=\frac{\mathbb{E}[F^2]}{(\mathbb{E}[F])^2}$ and a similar formula can be used for $V_{\rm rel}[G]$).
>
> $\textbf{ Question.}$ "Please consider comparison with Annealed Importance Sampling (AIS)  which samples a single transition at each temperature such that stationary samples at each temperature aren't needed.''
>
>
> $\textbf{Answer.}$
> Although AIS is similar to our work in that it uses a cooling schedule and MCMC, and it can be used to estimate the partition function of Gibss sampler,  there are three main differences between the two methods which prohibit a direct comparison.
>
> (1) The cooling schedule $\beta_0,\beta_1,\dots, \beta_n$ and its length in AIS are chosen heuristically, and they are fixed.
> However, in the TPA schedule the value of each $\beta_i$  depends on the previous sample $x_{i-1}$ drawn at temperature $\beta_{i-1}$ (it is an adaptive schedule).
>
>
> (2) As pointed out in section 4 of [N], the  efficiency of AIS  deteriorates when  the variance of the normalized importance weights,
> $\frac{V_g[w^{(i)}]}{\mathbb{E}_g[w^{(i)}]}$ is large, and we don't have general techniques to mathematically bound them.
>
>
> (3)   The TPA schedule is  specifically designed for paired product estimators, and it guarantees
> that $V_{\rm rel}[F]$ and $V_{\rm rel}[G]$ are constant (w.h.p.). This result only holds for paired product estimators $F$ and $G$, and
> does not hold for, e.g., the variance or relative variance of importance sampling weights. It is shown by [K] that under oracle access assumption the length of the TPA schedule is optimal. Note that the cooling schedule of AIS is set heuristically and often has longer length that the schedule obtained from TPA. Furthermore, there is no guarantee under these heuristics that $V_g[w^{(i)}]$s are bounded.
>
>
> We remark that finding rigorous bounds for AIS's performance and accuracy is an interesting research   direction,  but it requires developing tools and techniques different from ours.
>
>
>
> $\textbf{Question.}$
>  "If AIS can indeed be used in this scenario, the second "adaptation" can only be justified thorough empirical evaluation.''
>
> $\textbf{Answer.}$ here is why the second adaptive part of our algorithm cannot be replaced with the importance sampling method in AIS to achieve similar result to our analysis.
>
>
> Chapter 12 of [LPW], shows a relationship between variance of MCMC samples and the Markov chain's relaxation time (see, e.g., equations 12.2 and 12.8 of Chapter.12 [LPW]), which can be used to show that when the relaxation time of a Markov chain is large, the variance of importance weights can also be large for the worst-case initial distribution. AIS aims to resolve this issue by choosing (heuristically) a long schedule, so that the initial and desired distribution are close. However, since   theoretical analysis has not been provided for AIS, the variance of weights
> may be pessimistically large for pathological Gibbs samplers.
>
> The following example provides intuition for the relationship between the variance of the weights in MCMC importance sampling  and the Markov chain's relaxation time:
> Consider a Markov chain with state space \{$1,2$\} and  transition probabilities  $P(1,2)=P(2,1)=\epsilon$ and $P(1,1)=P(2,2)=1-\epsilon$. The stationary distribution of this chain is $(1/2,1/2)$ regardless of $\epsilon$, and its relaxation time  is $\frac{1}{2\epsilon}$. Thus as $\epsilon$ becomes smaller, the relaxation time becomes larger.
> Consider a simple scenario wherein importance sampling is done running only one step of MCMC. Any larger number of steps can be similarly shown using the results of Chapter.12 of [LPW]. The number of MCMC runs (each using one step) has to be taken proportional to $V[w^{(i)}]$, where $w^{(i)}(x)=\frac{\pi(x)}{P(\nu_0,x)}$, $P(\nu_0,x)$ is the probability of observing $x$ when starting at $\nu_0$ and $\pi$ is the desired (stationary) distribution.
> Having $\nu_0=(1,0)$ we have $P(\nu_0,1)= 1-\epsilon$  and
>  $P(\nu_0,2)= \epsilon$.
>  Thus $w^{(i)}(x=1)=\frac{1}{2(1-\epsilon)}$ and $w^{(i)}(x=2)=\frac{1}{2\epsilon}$.
>  Therefore,
>   $\mathbb{E}_g[w^{(i)}]= 1$ and $V_g[w^{(i)}]= (1-\epsilon)\left( \frac{1}{2(1-\epsilon)}-1 \right)^2+ \epsilon\left( \frac{1}{2(\epsilon)}-1 \right)^2\geq \epsilon\left( \frac{1}{2\epsilon}-1 \right)^2\simeq \frac{1}{2\epsilon}$.
>
>
>
> We finally note that the goal of our paper is to develop an algorithm with mathematically sound performance guarantees  and proven finite sample   complexities (and not heuristics). While such studies are ubiquitous in  i.i.d.  settings, they are rare in MCMC settings.  We feel that rigorous guarantees for MCMC settings are important as they are increasingly employed in real world machine learning applications, where safety, ethical, and legal concerns preclude the use of algorithms with only heuristic guarantees.
>
> [RR] Roberts, Gareth O., and Jeffrey S. Rosenthal. "Examples of adaptive MCMC." Journal of computational and graphical statistics 18.2 (2009): 349-367.
>
>
> [N] Neal, Radford M. "Annealed importance sampling." Statistics and computing 11.2 (2001): 125-139.
>
> [H]  Huber, Mark. "Approximation algorithms for the normalizing constant of Gibbs distributions." The Annals of Applied Probability 25.2 (2015): 974-985.
>
> [LPW] Levin, David A., and Yuval Peres. Markov chains and mixing times. Vol. 107. American Mathematical Soc., 2017.
>
> [K] Kolmogorov, Vladimir. "A faster approximation algorithm for the Gibbs partition function." COLT 2018.

---

> > ### Comment · Reviewer_DZFN · 2021-09-03
> > **Concerns addressed**
> >
> > The authors addressed most of my major concerns. I understand that the authors' method has better theoretical guarantees compared to AIS.
> > However I would highly recommend an empirical comparison which the above discussion does not preclude.
> > I totally agree that mathematical rigor is important. However fair comparisons to baselines are also equally important to the practitioner.
> > Based on the above fact and after reading other reviews regarding the clarity of exposition and subsequent responses I vote to maintain my rating.

---

### Decision · Program_Chairs · 2021-09-27

**Decision:**

Accept (Poster)

**Comment:**

The goal of this paper is a better algorithm to estimate the partition function / normalization constant in a Gibbs distribution. These are often of interest in many different forms in the NeurIPS community. The best current methods are based on annealing, essentially starting at a normalized distribution and moving through a series of distributions of different temperatures and estimating the ratio of each using MCMC. This paper proposes a new algorithm in this line which is intended to be less sensitive to tightness in the mixing times of MCMC.

This is a highly mathematical paper, where the goal is improved theoretical guarantees (rather than, say, practical performance improvement.) Unfortunately, all reviewers signaled that they were not confidently able to verify the correctness of the core results. In the end, the SAC stepped in and did a careful reading of the paper and recommends acceptance.

The SAC's reasoning is as follows:

The presentation is indeed terse and dense. But that is par for the course for theoretical papers at NeurIPS given the page limit. Everything was well defined (as far as I saw), and all the arguments seemed credible. The authors outlined the strategy of their arguments, hiding the considerable complexity of all the details in the supplementary material. I actually found the writing pretty good compared to the average published Neurips paper.

Regarding the concern by one reviewer about the assumption re spectral gap: I do not see this as a downside. It seems inevitable that one will need some assumptions on the dynamics of these chains in order to say anything theoretical. The goal (in my mind) is not to attempt the impossible task of theoretically explaining what "really" happens in practice, because one never knows. Assuming something about a spectral gap in the transition matrix of a markov chain seems to me to be less fraught epistemologically that (say) assuming one's data is drawn iid from some distribution ... an assumption that I doubt is true of almost all practical applications of machine learning, but upon which almost all ML theory is built... (Thus my view is aligned to that of the authors in their "Note to chair")

So it looks like this is a real advance on a problem of interest. The analysis looks credible (all one can ever really tell without spending a week working through every single step). It is easily at least as clear, correct, and significant as the majority of published Neurips papers so I do not have any qualms recommending acceptance....